# Combined ALK and MDM2 inhibition increases antitumor activity and overcomes resistance in human *ALK* mutant neuroblastoma cell lines and xenograft models

Hui Qin Wang[1†], Ensar Halilovic[1†], Xiaoyan Li[1], Jinsheng Liang[1], Yichen Cao[1], Daniel P Rakiec[1], David A Ruddy[1], Sebastien Jeay[2], Jens U Wuerthner[2], Noelito Timple[3], Shailaja Kasibhatla[3], Nanxin Li[3], Juliet A Williams[1], William R Sellers[1], Alan Huang[1], Fang Li[1]*

[1]Disease Area Oncology, Novartis Institutes for BioMedical Research, Cambridge, United States; [2]Disease Area Oncology, Novartis Institutes for BioMedical Research, Basel, Switzerland; [3]Genomics Institute of the Novartis Research Foundation, San Diego, United States

**Abstract** The efficacy of ALK inhibitors in patients with *ALK*-mutant neuroblastoma is limited, highlighting the need to improve their effectiveness in these patients. To this end, we sought to develop a combination strategy to enhance the antitumor activity of ALK inhibitor monotherapy in human neuroblastoma cell lines and xenograft models expressing activated ALK. Herein, we report that combined inhibition of ALK and MDM2 induced a complementary set of anti-proliferative and pro-apoptotic proteins. Consequently, this combination treatment synergistically inhibited proliferation of *TP53* wild-type neuroblastoma cells harboring *ALK* amplification or mutations in vitro, and resulted in complete and durable responses in neuroblastoma xenografts derived from these cells. We further demonstrate that concurrent inhibition of MDM2 and ALK was able to overcome ceritinib resistance conferred by MYCN upregulation in vitro and in vivo. Together, combined inhibition of ALK and MDM2 may provide an effective treatment for *TP53* wild-type neuroblastoma with *ALK* aberrations.

*For correspondence: fli@tangotx.com

†These authors contributed equally to this work

Competing interests: The authors declare that no competing interests exist.

## Introduction

Neuroblastoma is a common pediatric solid tumor that arises from neural crest cells, typically in the adrenal medulla or paraspinal ganglia with metastases to bone and bone marrow in high-risk cases (*Maris, 2010*). It remains a leading cause of childhood cancer-related death despite the use of multi-modal dose-intensive chemotherapy, radiation therapy and immunotherapeutic strategies (*Smith et al., 2010*). More effective therapeutic approaches are needed to improve cure rates among children with high-risk neuroblastoma while minimizing treatment-related toxicity. Recent efforts have been directed toward the development of new treatment strategies targeting validated molecular abnormalities underlying the malignant process.

Highly penetrant, heritable *ALK* mutations have been identified as the major cause of familial neuroblastoma. Somatic mutations in *ALK* are also found as oncogenic drivers in up to 10% of sporadic neuroblastoma with a gene amplification frequency of approximately 2% (*Chen et al., 2008*; *George et al., 2008*; *Janoueix-Lerosey et al., 2008*; *Mossé et al., 2008*). These mutations cause

single amino acid substitution in the ALK kinase domain and result in autophosphorylation and constitutive activation of the RTK. The most frequently mutated residues, R1275, F1174 and F1245, account for 85% of mutations in ALK (*Bresler et al., 2014*). The discovery of germline and somatic activating mutations in *ALK* provides a molecular rationale and a tractable target for treating neuroblastoma.

Crizotinib is a small-molecule adenosine triphosphate (ATP)-competitive inhibitor that has activity against ALK, MET and ROS1 RTKs (*Cui et al., 2011*). Therapy with crizotinib has significant clinical activity in patients with non-small cell lung cancer (NSCLC), anaplastic large cell lymphoma (ALCL) and inflammatory myofibroblastic tumor (IMT) that harbor *ALK* rearrangements (*Kwak et al., 2010*; *Mossé et al., 2013*). The marked clinical success of crizotinib in treating *ALK*-positive NSCLC tumors led to an early phase clinical trial in patients with recurrent or refractory neuroblastoma (*Mossé et al., 2013*). However, only one of 11 neuroblastoma patients with known *ALK* mutations responded in this study. Ceritinib is a second-generation ALK inhibitor that has 20-fold higher potency against ALK than crizotinib in enzymatic assays (*Marsilje et al., 2013*). It has demonstrated marked clinical activity in both crizotinib-naive and crizotinib-relapsed *ALK*-positive NSCLC patients (*Shaw et al., 2014*). In a phase I study of pediatric cancer, 14 neuroblastoma patients with known *ALK* mutations were treated with ceritinib (*Birgit Geoerger et al., 2015*). To date, only two patients showed partial responses, and one patient with ALK F1174L mutated neuroblastoma had shrinkage of a retroperitoneal mass. Overall, the responses of relapsed neuroblastoma with known *ALK*-activating mutations to crizotinib or ceritinib were not optimal for such targeted therapies.

ALK F1174L and F1174C mutations were identified as acquired resistance mutations in oncogenic ALK fusion proteins during crizotinib and ceritinib treatments, respectively (*Friboulet et al., 2014*; *Sasaki et al., 2010*). Biochemical analyses revealed that the F1174L mutation increased ATP-binding affinity of ALK and reduced its susceptibility to crizotinib inhibition (*Bresler et al., 2014*, *Bresler et al., 2011*). In a recent study, Infarinato and colleagues have shown that PF06463922, a highly potent ALK inhibitor for wild-type ALK and various ALK mutants, exhibits superior antitumor activity in neuroblastoma xenografts harboring the ALK F1174L mutation (*Infarinato et al., 2016*), providing a rationale for assessing the efficacy of newer generation ALK inhibitors in clinical trials for treating patients with *ALK*-mutated neuroblastoma. However, the majority of the tumors that harbor the crizotinib- and ceritinib-sensitive ALK R1275Q mutation did not respond in the trials of crizotinib and ceritinib in neuroblastoma patients (*Mossé et al., 2013*; *Birgit Geoerger et al., 2015*). These results suggest that the low response rates in these clinical studies is not likely to be solely attributed to the inherent primary resistance caused by the ALK F1174 substitutions in neuroblastoma. In addition, *ALK* mutations have been identified in both low- and high-risk neuroblastoma with equal frequency (*Bresler et al., 2014*), suggesting that activated ALK cooperates with other oncogenic aberrations to define high- versus low-risk tumors. For example, *ALK* mutations are frequently observed in *MYCN*-amplified neuroblastomas, and most of *ALK* amplifications co-occur with *MYCN* amplification (*George et al., 2008*; *Bresler et al., 2014*; *Bagci et al., 2012*; *De Brouwer et al., 2010*). ALK mutations that co-occur with *MYCN* amplification are biased toward F1174 substitutions. Constitutively, activated ALK synergizes with MYCN overexpression in inducing neuroblastoma in animal models, and the co-occurrence of ALK F1174 mutations and *MYCN* amplification defines a subset of neuroblastoma patients with particularly poor outcome (*Berry et al., 2012*; *Heukamp et al., 2012*; *Zhu et al., 2012*). Therefore, the co-occurrence of mutations in *ALK* with dysregulation in other oncogenic drivers, such as *MYCN* amplification, may further limit the activity of single-agent ALK inhibitors.

Combinatorial therapies that target other signaling pathways in addition to ALK may be required to improve the effectiveness of ALK inhibitors in neuroblastomas that harbor *ALK* aberrations. In this study, we assessed the antitumor activity of ceritinib in combination with NVP-CGM097, a potent and selective small molecule inhibitor of MDM2, in *ALK*-mutated or *ALK*-amplified neuroblastoma models. MDM2 functions as an E3 ubiquitin ligase that targets p53 for proteasome-mediated degradation. p53 protein plays a critical role in tumor suppression by regulating the expression of genes involved in cell cycle arrest and apoptosis. Thus, CGM097 exerts its antitumor effects by stabilizing p53 and activating p53 pathways (*Jeay et al., 2015*; *Weisberg et al., 2015*). However, to elicit these responses, tumor cells must possess wild-type or functional p53 (*Jeay et al., 2015*; *Weisberg et al., 2015*). In contrast to the high frequency of *TP53* mutations observed in many human cancers of adults, mutations of *TP53* have been reported in less than 2% of neuroblastomas at diagnosis and

15% at relapse (*Carr-Wilkinson et al., 2010*; *Tweddle et al., 2003*). Here, we report that the combination of ceritinib with CGM097 promotes apoptosis in *ALK* mutant/*TP53* wild-type neuroblastoma cell lines and results in complete tumor regression and markedly prolonged survival in neuroblastoma xenograft models. In addition, ceritinib and CGM097 combination overcomes acquired ceritinib resistance caused by MYCN upregulation in an ALK-driven neuroblastoma model. Our study as well as the exceptionally low rate of *TP53* mutations in neuroblastoma provides the rationale for testing combinatorial inhibition of ALK and MDM2 as a therapeutic approach for treating *TP53* wild-type neuroblastomas with aberrantly activated ALK.

## Results

### Treatment of single-agent ALK inhibitors is not sufficient for maximal antitumor effect in neuroblastoma models expressing constitutively activated ALK

We first examined the anti-proliferative and cytotoxic effect of four ALK inhibitors – crizotinib, ceritinib, alectinib and PF06463922 – in four neuroblastoma cell lines that harbor *ALK* aberrations. These cell lines were characterized with respect to their genetic status of *ALK*, *MYCN* and *TP53* (*Table 1*). Crizotinib, ceritinib and alectinib have been approved by multiple health authorities to treat advanced NSCLC harboring ALK rearrangements, whereas PF06463922 is a next-generation ALK inhibitor with higher potency and selectivity and appears to be active against all known clinically acquired ALK mutations (*Johnson et al., 2014*; *Zou et al., 2015*). Zou *et al.* have directly compared the potency of crizotinib, ceritinib, alectinib and PF06463922 in Ba/F3 cells engineered to express wild-type or F1174L-mutated EML4-ALK (*Zou et al., 2015*). They reported that the $IC_{50}$ values of crizotinib, ceritinib, alectinib and PF06463922 on cell proliferation of Ba/F3 cells expressing wild-type EML4-ALK were 88 nM, 25 nM, 39 nM and 2.7 nM, respectively; and 106 nM, 40 nM, 28 nM and 4.8 nM, respectively, in Ba/F3 cells expressing EML4-ALK F1174L mutant.

NB-1 cells, which carry *ALK* amplification and no *ALK* mutations in the kinase domain, displayed pronounced dose-dependent decrease in cell viability when treated with the four ALK inhibitors (*Figure 1A*). The sensitivity of NB-1 matches the potency of these ALK inhibitors in Ba/F3 cells expressing wild-type EML4-ALK as reported by *Zou et al. (201)5* (*Figure 1B*) and was reflected by the levels of inhibition of phospho-ALK and the components of the downstream signaling of ALK (*Figure 1C*). We next assessed the activity of ceritinib against xenografts derived from NB-1 cells. The response of NB-1 xenografts to ceritinib was transient, and tumors resumed growth 2 weeks after the start of the treatment (Figure 3A and *Figure 3—source data 1*), suggesting that single-agent ALK inhibition may not be sufficient to achieve durable response in a subset of ALK-driven neuroblastoma tumors.

All these four ALK inhibitors displayed reduced anti-proliferative activity in the neuroblastoma cell lines that harbor the ALK F1174L mutation compared with NB-1 (*Figure 1A and B*). Although PF06463922 most robustly inhibited phospho-ALK and phospho-ERK in these cell lines (*Figure 1C*), we did not observe significantly greater growth inhibition in the F1174L-mutated cell lines by

**Table 1.** *ALK*-amplified, *ALK*-mutant and ALK wild-type neuroblastoma cell lines of varying *TP53* and *MYCN* status.

| Cell line | ALK status | MYCN status | TP53 status |
|---|---|---|---|
| NB-1 | Amp | Amp | Wt |
| SH-SY5Y | F1174L | Non-Amp | Wt |
| KP-N-RT-BM-1 | F1174L | Amp | Wt |
| NB-1643 | R1275Q | Amp | Wt |
| IMR-32 | Wt | Amp | Wt |
| KELLY | F1174L | Amp | Mut |

Amp, amplified; Non-Amp, non-amplified; Wt, wild type; Mut, mutant

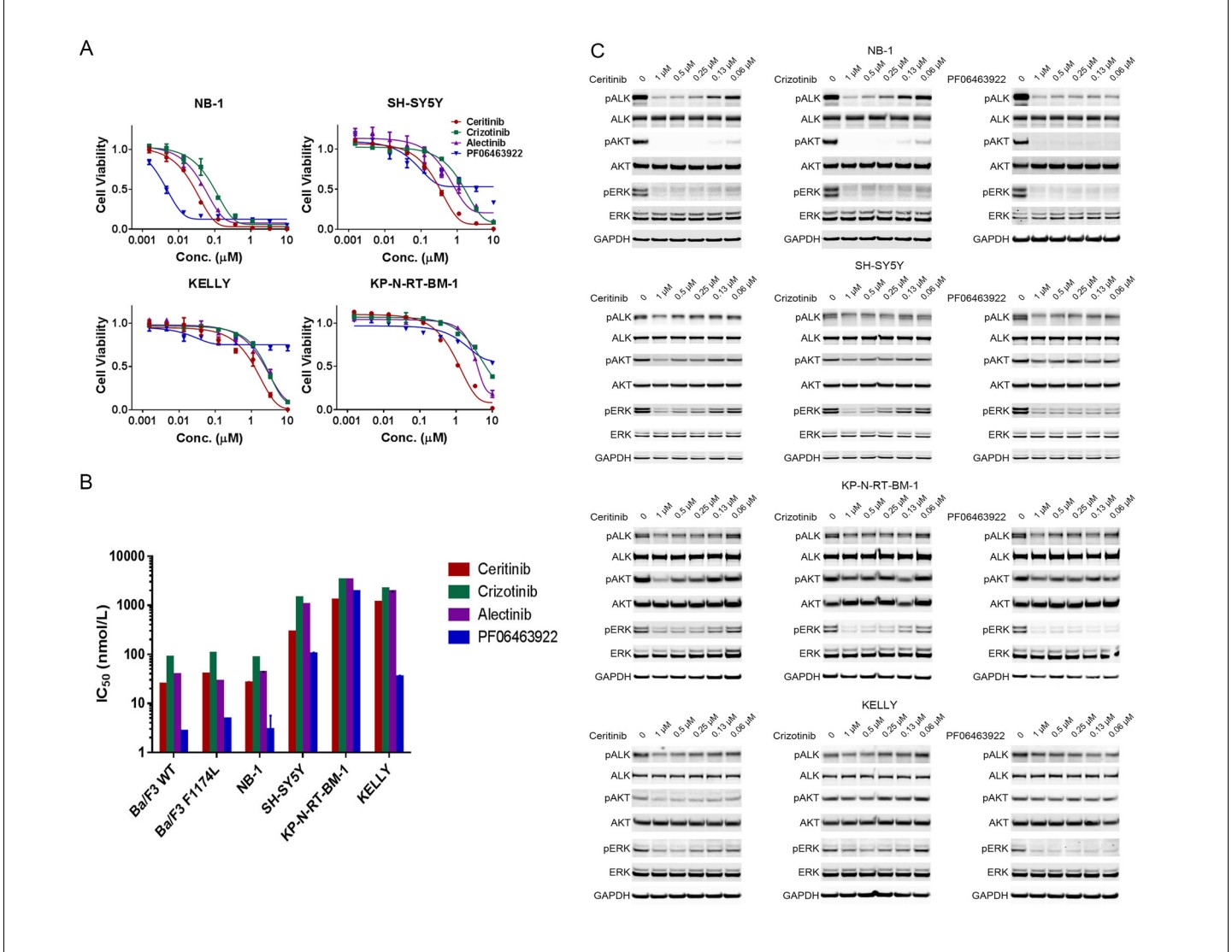

**Figure 1.** ALK inhibitors inhibit or partially inhibit the proliferation of neuroblastoma cell lines harboring ALK aberrations. (**A**) Dose response curves of *ALK*-amplified or *ALK* mutant neuroblastoma cell lines exposed to the ALK inhibitors ceritinib, crizotinib, alectinib and PF06463922. Relative cell growth was quantified using CellTiter-Glo on day 6. Data are shown as mean ± SEM from three biological replicates. (**B**) IC50 values of ceritinib, crizotinib, alectinib and PF06463922 across Ba/F3 cells expressing wild-type or F1174L-mutated EML4-ALK and neuroblastoma cell lines harboring ALK aberrations. The data of Ba/F3 cells were reported by Zou and colleagues (28). (**C**) Inhibition of ALK autophosphorylation and downstream signaling by ceritinib, crizotinib and PF06463922 in *ALK*-amplified or *ALK* mutant neuroblastoma cell lines. Cells were harvested after treatment for 4 hr with the indicated compounds at different concentrations. Whole cell lysates were analyzed by Western blotting to detect the levels of ALK, AKT and ERK proteins and their phosphorylation.

PF06463922 compared with crizotinib, ceritinib and alectinib (*Figure 1A*). Consistent with prior observations (*Infarinato et al., 2016*), PF06463922 caused a partial growth inhibition in cell lines with the ALK F1174L mutation (*Figure 1A*), presumably due to the high specificity of PF06463922 for ALK. In addition, the IC$_{50}$ values for the four ALK inhibitors in the F1174L neuroblastoma cell lines were 10 to 100-fold higher than the IC$_{50}$ values in Ba/F3 cells expressing EML4-ALK F1174L mutant (*Zou et al., 2015*) (*Figure 1B*). These data suggest that, in addition to ALK signaling, other pathways may contribute to the survival of the F1174L-mutated neuroblastoma cells. To further assess the activity of ceritinib against neuroblastoma tumors that harbor the F1174L mutation in vivo, we examined the antitumor effect of ceritinib in SH-SY5Y xenograft tumors in mice. Treatment of ceritinib

showed limited efficacy in SH-SY5Y xenograft tumors, resulting in slower tumor growth compared with vehicle-treated mice (Figure 3A and *Figure 3—source data 1*). Altogether, these data suggest that ALK inhibition alone is not sufficient to achieve durable and deep responses in neuroblastoma tumors that contain ALK aberrations, and that a combination therapy that consists of an ALK inhibitor and a second agent is required.

## Concurrent activation of the tumor suppressor p53 and inhibition of ALK synergistically suppress neuroblastoma growth

Small-molecule MDM2 antagonists that disrupt the interaction between p53 and MDM2 to activate the p53 pathway have shown initial signs of antitumor efficacy in patients with solid and hematologic malignancies (*Andreeff et al., 2016*; *Ray-Coquard et al., 2012*). MDM2 antagonists function through activation of p53 and hence efficacy of this therapeutic class is restricted to tumors that retain wild-type *TP53*. Mutations in *TP53* occur rarely in neuroblastoma (*Tweddle et al., 2003*), and p53 pathway is mainly inactivated through mechanisms that affect stability and nuclear shuttling of p53 protein, such as MDM2 amplification and p14ARF impairment (*Carr-Wilkinson et al., 2010*). These tumors should remain amenable to MDM2 inhibition. In order to test this, we examined the anti-proliferative effect of the MDM2 inhibitor CGM097 in a panel of neuroblastoma cell lines, including five *TP53* mutant and six *TP53* wild-type cell lines (*Figure 2A*). Consistent with the mechanism of action of MDM2 inhibitors, the *TP53* wild-type neuroblastoma cell lines were significantly more sensitive than the *TP53* mutant cell lines. Whereas all five *TP53* mutant cell lines had $IC_{50}$ values greater than 11 µM, the $IC_{50}$ values of all the *TP53* wild-type cell lines were below 3.5 µM (range 0.8–3.5 µM) (*Figure 2A*). To evaluate the in vivo antitumor activity of CGM097 against *TP53* wild-type neuroblastoma tumors that harbor wild-type *ALK* or *ALK* aberrations, we treated xenografts derived from *ALK*-amplified NB-1 cells, F1174L-mutated SH-SY5Y cells, R1275Q-mutated NB-1643 cells and *ALK* wild-type IMR-32 cells. CGM097 alone at 50 mg/kg/day demonstrated no inhibition of tumor growth in these models (Figure 3A and *Figure 3—source data 1*), suggesting that single-agent treatment with MDM2 inhibitors is not sufficient to inhibit the growth of *TP53* wild-type neuroblastoma tumors expressing wild-type or activated ALK.

The identification of *ALK*-activating mutations in a subset of neuroblastoma tumors, the extremely low mutation rate of *TP53* and the insufficiency of single-agent ALK or MDM2 inhibitors to inhibit the growth of this type of tumors prompted us to hypothesize that inhibition of the ALK pathway with concurrent induction of the p53 pathway might result in synergistic antitumor effects. To test this hypothesis, we exposed the four neuroblastoma cell lines that harbor *ALK* aberrations to combinations of ceritinib and CGM097. Of these four cell lines, one carried *TP53* mutation (KELLY), and the rest were *TP53* wild-type (*Table 1*). Cells were exposed to ceritinib and CGM097 at concentrations ranging from 0 to 3.3 µM and cell proliferation was measured at 72 hr. The 64 different combinations for each cell line are depicted in a dose matrix in which percentages of growth inhibition relative to non-treated cells are indicated and visualized using a color scale (*Figure 2B*). A combination index that measures dose shifting when combination treatments achieve a 50% growth inhibitory effect was calculated from the isobologram for each cell line to represent the strength of synergy (*Lehár et al., 2009*). An examination of these matrices indicates that the combinations of ceritinib and CGM097 at the concentrations greater than 100 nM resulted in improved growth inhibition compared to treatments with either compound alone in all three *TP53* wild-type cell lines (*Figure 2B*). Isobologram analysis revealed that the effects of combining ceritinib and CGM097 were synergistic in these three cell lines (*Figure 2C*). In contrast to the *TP53* wild-type cell lines, CGM097 alone was completely ineffective in inhibiting the growth of the *TP53*-mutated KELLY cells, and no improved growth inhibition of the combination was observed in these cells (*Figure 2B and C*).

To elucidate the molecular mechanism of synergy between the ALK and MDM2 inhibitor, we examined their effects on protein signaling and pathway modulation. Effective target inhibition was achieved with CGM097, as evident by an accumulation in p53 protein itself and the induction of p53 direct effectors, p21 and MDM2 proteins (*Figure 2D*). Ceritinib caused a reduction of phospho-ALK, phospho-AKT and phospho-ERK signaling, likewise demonstrating effective target and pathway inhibition (*Figure 2D*). As expected, CGM097 was unable to induce p53 protein or its downstream effectors in *TP53* mutant KELLY cells as expected (*Figure 2D*). When the components in the cell cycle and apoptosis pathways were examined, CGM097 caused induction of p21 and PUMA, while ceritinib caused induction of p27 and Bim (*Figure 2D*). Therefore, together ceritinib and CGM097

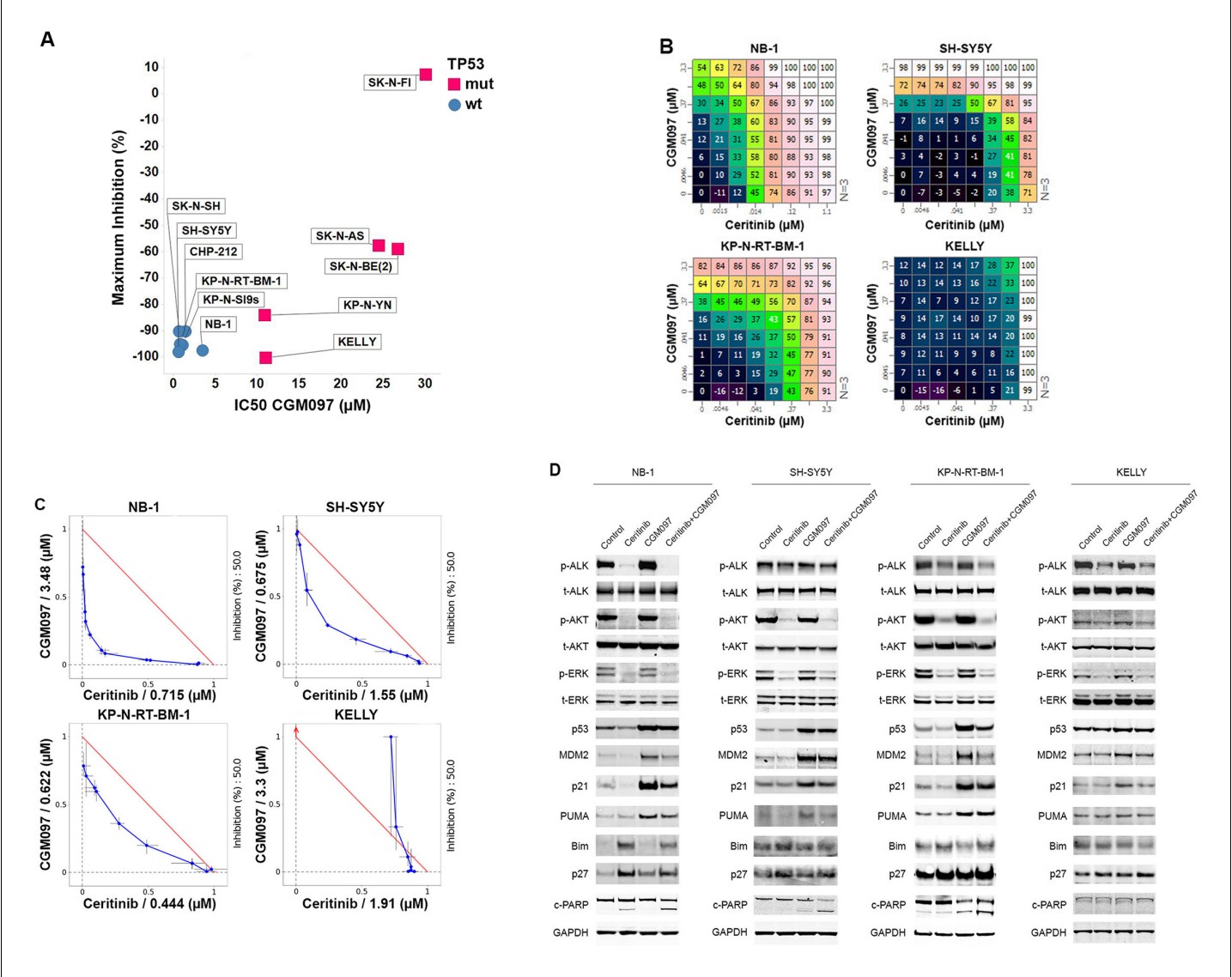

**Figure 2.** Combination of ceritinib with CGM097 leads to increased antitumor activity in *TP53* wild-type neuroblastoma cell lines harboring *ALK* aberrations. (A) Sensitivity of a panel of *TP53* wild-type and mutant neuroblastoma cell lines to CGM097 treatment for 72 hr. Maximum percentages of inhibition are on the y-axis, and IC50 values are on the x-axis. (B) Growth effects of combining ceritinib and CGM097 in *ALK*-amplified or *ALK* mutant neuroblastoma cells. In each grid, the effects correspond to increasing doses of CGM097 on the y-axis and increasing doses of ceritinib on the x-axis. All remaining points on the grid display the results of the combination of the two inhibitors that correspond to the single-agent concentrations denoted on the two axes. Values are displayed as percentage of inhibition based on comparisons made with the Day 3 untreated controls. (C) Isobologram analysis of the data in (B). Doses of CGM097 are on the y-axis, and ceritinib on the x-axis. The red straight line in each panel defines all the pairs of doses of ceritinib and CGM097 that lead to 50% of proliferation inhibition from simple additivity. The points on the blue curve represent the actual doses of ceritinib and CGM097, when combined, to achieve 50% of inhibition. The blue curves of the *TP53* wild-type cell lines bow under the red lines, indicating the combination of ceritinib and CGM097 is synergistic. (D) Enhanced apoptosis as evidenced by increased levels of cleaved PARP in *TP53* wild-type neuroblastoma cells with *ALK* amplification or mutation when treated with ceritinib in combination with CGM097. Cells were incubated with DMSO, 1 µM ceritinib, 2 µM CGM097 and 1 µM ceritinib plus 2 µM CGM097 for 16 hr.

induced a complementary, yet non-overlapping set of anti-proliferative and apoptosis-stimulating molecules, which resulted in synergistic antitumor effects (*Figure 2D*). Only when combined, the two compounds could cause significant increase of cleaved PARP (*Figure 2D*), suggesting that the combination leads to cell death.

Given the improved in vitro efficacy of combined ALK and MDM2 inhibition, we next determined whether ceritinib in combination with CGM097 was effective in the neuroblastoma mouse xenograft models. Mice bearing tumor xenografts were treated with vehicle, ceritinib, CGM097 or both. CGM097 was not effective in NB-1 (*ALK* amplified), SH-SY5Y (ALK F1174L) and NB-1643 (ALK R1275Q) models (*Figure 3A* and *Figure 3—source data 1*). In ceritinib single-agent treated group, NB-1 tumors exhibited an initial shrinkage, but resumed growth quickly (*Figure 3A* and *Figure 3—source data 1*). Ceritinib monotherapy was partially effective in the SH-SY5Y and NB-1643 models, resulting in slower tumor growth compared with the vehicle-treated group (*Figure 3A* and *Figure 3—source data 1*). In contrast to the limited or transient responses to the single-agent treatments, the combination led to complete and durable tumor regressions throughout treatment period in NB-1, SH-SY5Y and NB-1643 models (*Figure 3A* and *Figure 3—source data 1*). Two (NB-1) or eight (SH-SY5Y) weeks after the combination treatment was stopped, two out of five animals and three out of four animals showed tumor regrowth for NB-1 and SH-SY5Y, respectively (*Figure 3A* and *Figure 3—source data 1*). The recurrent tumors remained sensitive to the combination treatment (data not shown), suggesting that the residual disease was not completely eliminated in a subset of animals. Neither single agent nor combination treatments showed antitumor activity in the IMR-32 xenograft tumors that contain wild-type *ALK* and *TP53* (*Figure 3A* and *Figure 3—source data 1*), suggesting that the combined inhibition of ALK and MDM2 provides an effective treatment in *TP53* wild-type neuroblastoma with *ALK* aberrations. The in vivo administration of ceritinib in combination with CGM097 for 3 days resulted in inhibition of phospho-ALK, induction of p21 and increased levels of cleaved PARP compared with either single agent alone (*Figure 3B*), which is in agreement with the observations made in vitro.

## The transcription of genes encoding ribosomal proteins and the protein level of MYCN are increased in NB-1 ceritinib-resistant tumors

Although tumors in ceritinib single-agent treated NB-1 mouse xenografts exhibited initial regressions, these tumors quickly resumed growth (*Figure 3A* and *Figure 3—source data 1*). To identify the molecular mechanism that allowed these tumors to grow out in the presence of continued ALK inhibition, we sequenced poly(A)-selected RNA extracted from tissues of NB-1 tumors that progressed on ceritinib treatment and those treated with a vehicle to identify differentially expressed genes between these two groups. We found a marked increase in the expression of a group of genes in the ceritinib-resistant tumors when we plotted the gene expression levels of the ceritinib-resistant tumors versus those of the vehicle-treated tumors (*Figure 4A* and *Supplementary file 1*). Interestingly, the majority of these genes upregulated in the ceritinib-resistant tumors encode RPS and RPL proteins of the small and large ribosomal subunits and translation elongation factors (*Figure 4A*). The Myc family of transcription factors stimulates ribosome biogenesis by enhancing the transcription of ribosomal RNA and genes encoding many RPS and RPL proteins (*van Riggelen et al., 2010*). In addition, MYCN has been shown to upregulate the transcription of RPS and RPL genes and translation initiation and elongation factors in human neuroblastoma cell lines (*Boon et al., 2001*). With the selective upregulation of genes encoding proteins of the small and large ribosomal subunits in the ceritinib resistant tumors, we hypothesized that the expression of MYCN might be increased in these tumors. Western blot analysis of three individual tumors collected from the vehicle-treated or ceritinib-resistant mice revealed an increase in the expression of MYCN in the NB-1 ceritinib-resistant tumors (*Figure 4B*).

## Increase in MYCN expression causes resistance to ceritinib in the NB-1 cells and CGM097 restores their sensitivity to ceritinib

To further investigate the role of MYCN upregulation in resistance to ceritinib, we exposed the NB-1 cells to 0.2 µM ceritinib in vitro. Initially, a large portion of the cells was killed or showed no signs of proliferation. After 3 weeks of treatment, the cells grew out in the presence of 0.2 µM ceritinib. The resultant cell line, NB-1R, showed significantly decreased sensitivity to ceritinib as compared to the parental cell line (*Figure 5A*). Similar to the observations made in the ceritinib-resistant NB-1 xenografts, Western blot analysis of these ceritinib-resistant cells revealed markedly increased expression of MYCN (*Figure 5B*). However, ALK signaling was strongly suppressed in the resistant cells, as exemplified by reduced levels of phospho-ALK, phospho-AKT and phospho-ERK (*Figure 5B*). We

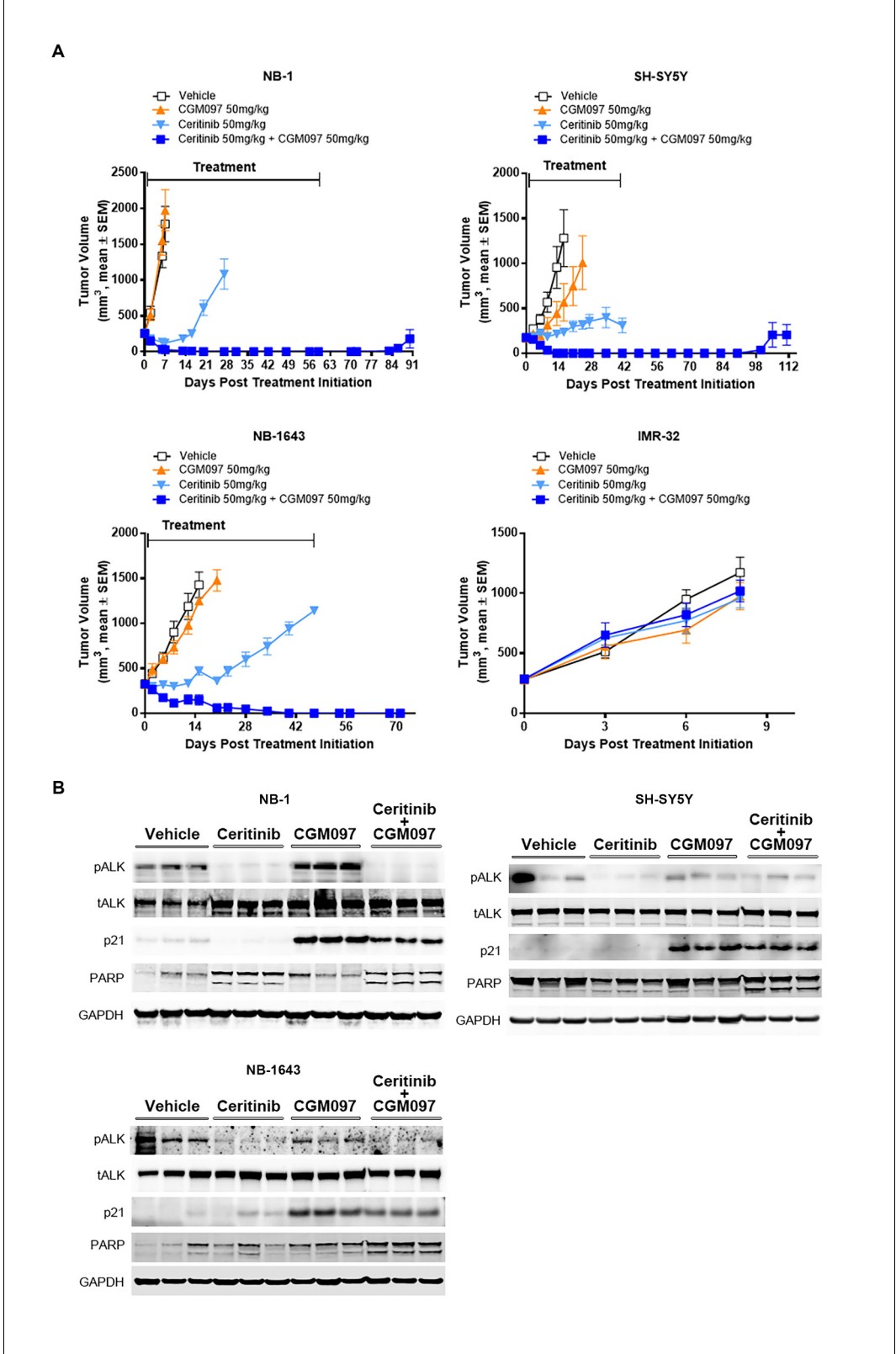

**Figure 3.** Combination of ceritinib with CGM097 leads to increased antitumor activity in *TP53* wild-type neuroblastoma xenograft tumors harboring *ALK* aberrations. (**A**) The improved in vivo efficacy in the NB-1, SH-SY5Y and NB-1643 xenograft mouse models that harbor wild-type *TP53* and *ALK* aberrations when ceritinib was combined with CGM097 and lack of antitumor activity of this combination in the IMR-32 xenograft tumors that harbor wild-type *TP53* and *ALK*. The neuroblastoma cell lines NB-1, SH-SY5Y and IMR-32 were implanted into the flanks of nude mice and NB-1643 in SCID

*Figure 3 continued on next page*

*Figure 3 continued*

mice. Animals were randomized into four groups when the average tumor volume was 200–300 mm$^3$ and received vehicle, ceritinib (50 mg/kg), CGM097 (50 mg/kg) or both inhibitors in combination. Combination of ceritinib with CGM097 was withdrawn on day 60 and day 45 for NB-1 and SH-SY5Y, respectively, to allow tumor regrowth. Tumor dimensions and body weights were measured at the time of randomization and twice weekly thereafter for the study duration. Average tumor volume and SEM are shown as a function of time. (B) Inhibition of phospho-ALK, induction of p21 and increased levels of cleaved PARP in NB-1, SH-SY5Y and NB-1643 xenograft tissues treated with ceritinib in combination with CGM097. Animals were treated with vehicle, ceritinib (50 mg/kg), CGM097 (50 mg/kg) or both inhibitors in combination for 3 days. Tumor tissues were recovered 4 hr after the last dose treatment and analyzed by Western blotting.

The following source data is available for figure 3:

**Source data 1.** Details of human neuroblastoma cell line xenograft studies.

further developed PF06463922-resistant SH-SY5Y and KP-N-RT-BM-1 cells by treating the parental cells with 1 µM of PF06463922 for 3 to 4 months (*Figure 5—figure supplement 1A*). To achieve adequate ALK inhibition while avoiding the off-target effects associated with high concentrations of ceritinib, we chose PF06463922 as the ALK inhibitor for these two cell lines because it is more potent against the ALK F1174L mutation than ceritinib. The levels of MYCN were again markedly increased in these two PF06463922-resistant cell lines compared to their parental cells (*Figure 5—figure supplement 1B*). These results suggest that despite the ALK inhibitor-mediated suppression of ALK signaling, MYCN upregulation may be sufficient to drive proliferation in the presence of ALK inhibitors.

To test whether upregulation of MYCN is responsible for ceritinib resistance, the NB-1R cell line was stably transduced with lentiviral doxycycline-inducible vectors containing non-target control sequence or shRNA against *MYCN*. In the absence of doxycycline, no reduction of MYCN protein was observed in the *MYCN* shRNA-expressing NB-1R cells (*Figure 5C*). In contrast, exposure to doxycycline led to a dose-dependent reduction in MYCN protein levels (*Figure 5C*). While MYCN knockdown in DMSO-treated NB-1R cells did not significantly affect the cell viability, downregulation of MYCN restored the sensitivity of the NB-1R cells to ceritinib (*Figure 5D*), suggesting that both ALK inhibition and reduction in MYCN level are required for inhibiting the growth of NB-1R cells. In addition, shRNA-mediated downregulation of MYCN resulted in increased levels of phospho-ALK but had no effect on the expression of p53 and p21 (*Figure 5—figure supplement 2*).

The contradictory roles of the MYC family in promoting cell proliferation and sensitizing cells to apoptosis have been well established (*Meyer and Penn, 2008*). MYC deregulation activates the p53 pathway and triggers apoptosis, which serves as a major mechanism for preventing MYC-mediated tumorigenesis. Tumors are able to evade MYC-induced apoptosis by introducing defects in the p14ARF-MDM2-p53 axis and its effectors and regulators, and restoring the abrogated p53 pathway may provide an approach to force MYC-deregulated tumors to undergo apoptosis. In the absence of *TP53* mutations, defects in the p53 pathway, in general, allow neuroblastoma cells to circumvent the safeguard against MYCN-driven neoplasia. Our RNA-Seq data revealed that *TP53* gene remained wild-type in the ceritinib-resistant NB-1R cells, as in the parental NB-1 cells. However, expression of p21, one of the major p53 transcriptional target genes, was strongly decreased in the ceritinib-resistant cells (*Figure 5E*), indicating that the p53 signaling might be impaired in the NB-1R cells.

To determine if activating the p53 pathway and increasing the expression of p21 could restore the sensitivity to ceritinib, we treated the NB-1R cells with CGM097 alone or in combination with ceritinib for 16 hr. CGM097 treatment resulted in an increase in p53 expression and induction of its target genes, including p21, MDM2 and PUMA (*Figure 5F*). In addition, CGM097 restored the sensitivity of the NB-1R, SH-SY5YR and KP-N-RT-BM-1R cells to ceritinib or PF06463922 to a level comparable to that in the parental cells (*Figure 5G* and *Figure 5—figure supplement 1C*). Ceritinib in combination with CGM097 caused increased anti-proliferative activity and decreased viability of NB-1R cells compared to either agent alone (*Figure 5G*), presumably through induction of stronger apoptosis, evident by induction of cleaved PARP (*Figure 5F*).

To further assess the activity of combination of ceritinib with CGM097 in overcoming the acquired resistance to ceritinib monotherapy in vivo, we treated NB-1 tumor-bearing mice with vehicle,

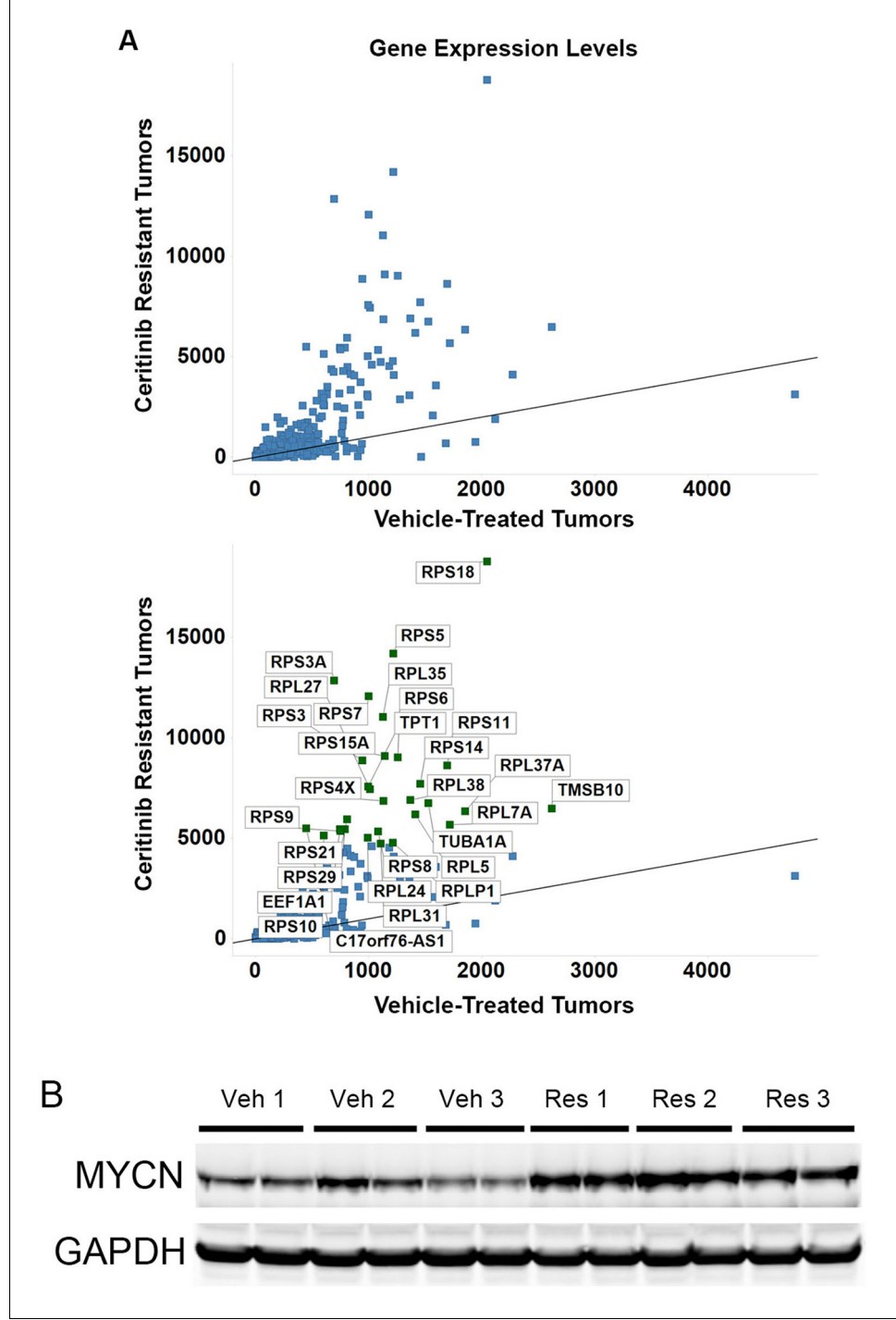

**Figure 4.** The transcription of ribosomal proteins and the protein level of MYCN are increased in NB-1 ceritinib-resistant tumors. (**A**) Upregulation of transcription of ribosomal protein components in NB-1 ceritinib-resistant tumors. RPS and RPL genes encode proteins of the small and large ribosomal subunits, respectively. The lower panel is identical to the upper panel except that the upregulated genes are labeled with their gene symbols. NB-1 tumors that progressed under treatment with 50 mg/kg ceritinib and NB-1 tumors treated with vehicle as shown in *Figure 3A* were analyzed by RNA-Seq. Gene expression levels quantified by FPKM are shown as the average from two individual tumors in each group. FPKM, Fragments Per Kilobase of transcript per Million fragments mapped. (**B**) Upregulation of MYCN protein in NB-1 ceritinib-resistant tumors. Tissues of NB-1 tumors that progressed during treatment with 50 mg/kg ceritinib and NB-1 tumors treated with vehicle as shown in *Figure 3A* were recovered and analyzed by Western blot analysis. Veh, vehicle-treated tumors. Res, ceritinib-resistant tumors.

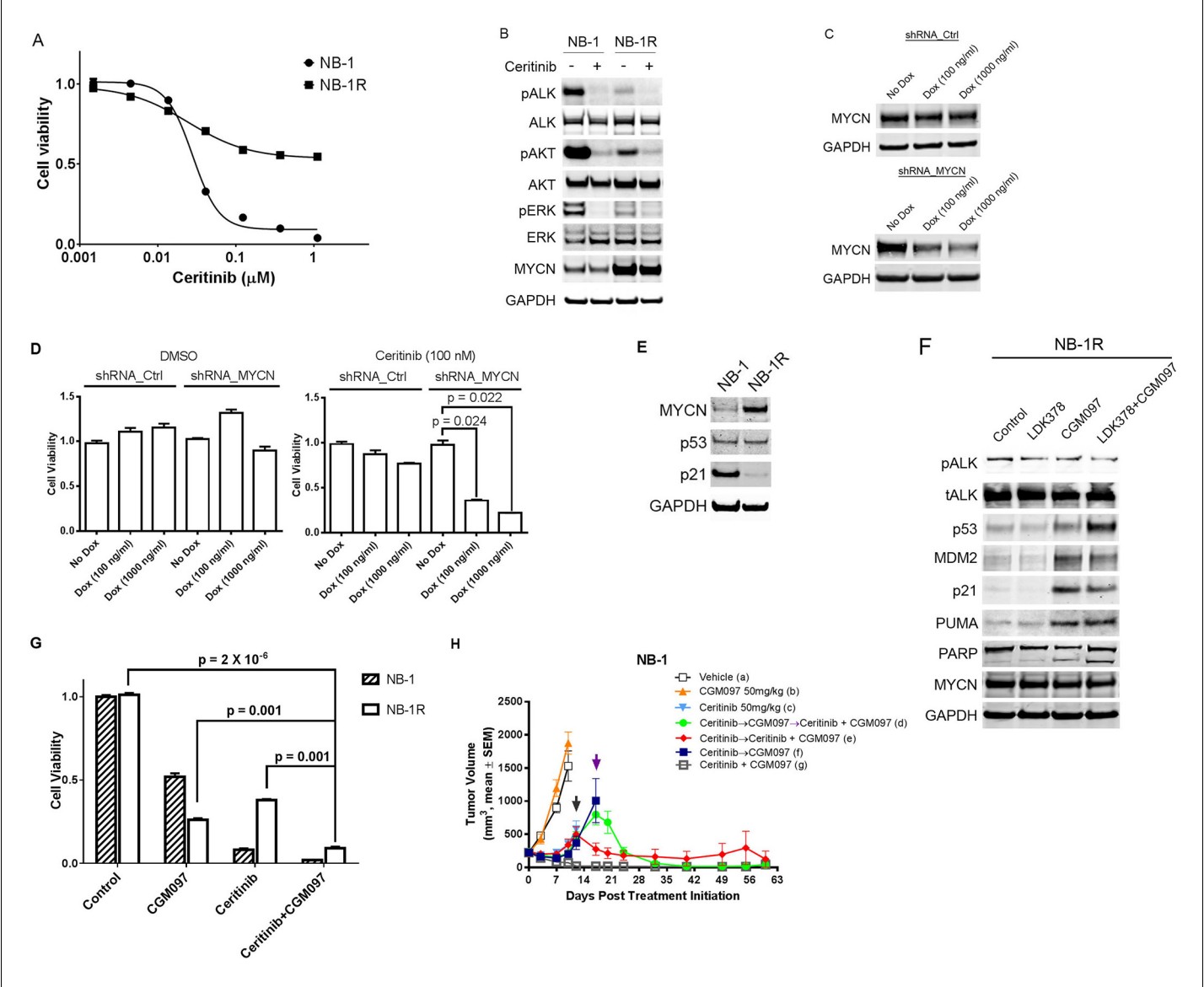

**Figure 5.** CGM097 restores ceritinib sensitivity in NB-1 cells that develop resistance to ceritinib as a result of increased level of MYCN. (**A**) Decreased sensitivity of NB-1R cells to ceritinib in vitro. NB-1R cells were derived from NB-1 cells by growing them in 200 nM ceritinib for 3 weeks. (**B**) Upregulation of MYCN in NB-1R cells. NB-1 parental and resistant cells were treated with ceritinib for 2 hr, and cell extracts were immunoblotted to detect the indicated proteins. (**C**) Downregulation of MYCN by a doxycycline-inducible MYCN-specific shRNA in NB-1R cells. Lentiviral doxycycline-inducible vectors containing non-target control sequence or shRNA against MYCN were introduced into NB-1R cells. The cells were exposed to doxycycline at increased concentrations for 7 days, and cell extracts were immunoblotted with MYCN antibodies. Dox, Doxycycline. (**D**) Restored sensitivity of NB-1R cells to ceritinib by knocking down the expression of MYCN. Doxycycline-inducible control and MYCN-specific shRNAs were introduced into NB-1R cells. The cells were treated with DMSO or 0.2 μM ceritinib in the presence or absence of doxycycline for 14 days. Relative cell growth was quantified using the CellTiter-Glo assay. Data are shown as mean ± SEM from three biological replicates. (**E**) Downregulation of p21 in NB-1R cells. Cell lysates of parental and resistant cells were immunoblotted to detect the indicated proteins. (**F**) Restored level of p21 in NB-1R cells treated with CGM097. NB-1R cells were treated for 16 hr with 0.2 μM ceritinib alone, 1 μM CGM097 alone, or the two compounds in combination. Cells were harvested after the treatments, and cell lysates were analyzed by Western blotting. (**G**) Increased anti-proliferative activity of ceritinib in combination with CGM097 in NB-1R cells. NB-1 parental and NB-1R cells were treated with 0.2 μM ceritinib alone, 1 μM CGM097 alone, or the two compounds in combination for 6 days. Relative cell growth was quantified using the CellTiter-Glo assay. Data are shown as mean ± SEM from three biological replicates. (**H**) Restored ability of ceritinib to inhibit ceritinib-resistant NB-1 tumor growth when it was combined with CGM097 in vivo. Animals were randomized into treatment groups following NB-1 cell implantation when the average tumor volume was ~ 220–250 mm3, and treatments were initiated. On day 10, mice in vehicle and CGM097-treated groups were taken down. On day 12, tumors in mice treated with 50 mg/kg ceritinib progressed, and they were randomized into the following groups: (1) ceritinib treatment was terminated and CGM097 treatment was started at the time point indicated by black arrow (dark blue curve), (2) ceritinib treatment was terminated and CGM097 treatment was started at the time point indicated by black arrow, and then treatment with

*Figure 5 continued on next page*

*Figure 5 continued*

ceritinib and CGM097 combination was started at the time point indicated by purple arrow (green curve), (3) ceritinib treatment was terminated and ceritinib and CGM097 combination treatment was started at the time point indicated by black arrow (red curve). All these treatments ended on day 60. Average tumor volume and SEM are shown as a function of time.

The following source data and figure supplements are available for figure 5:

**Source data 1.** Details of human NB-1 cell line xenograft studies.
**Figure supplement 1.** MYCN is upregulated in SH-SY5Y and KP-N-RT-BM-1 cells that are resistant to PF06463922 and CGM097 overcomes PF06463922 resistance.
**Figure supplement 2.** Downregulation of MYCN increases the level of phospho-ALK in NB-1R cells.

ceritinib alone, CGM097 alone or ceritinib plus CGM097. When the animals in the ceritinib-treated group became resistant to ceritinib (tumor sizes reached 400–500 mm$^3$), they were randomized again into two groups and the treatments were switched to either CGM097 monotherapy or combination of ceritinib with CGM097. Under CGM097 monotherapy treatment, tumors continued to grow (*Figure 5H*). In contrast, the combination treatment led to fast tumor regressions (red curve in *Figure 5H*). The ceritinib-resistant tumors that switched to CGM097 monotherapy continued to grow to 800–900 mm$^3$ and were re-challenged with the combination of ceritinib and CGM097. The combination treatment once again caused complete tumor regressions in these tumors (green curve in *Figure 5H*). CGM097 alone could not reverse the outgrowth of ceritinib-resistant tumors (dark blue curve in *Figure 5H*), indicating that both ALK inhibition and p53 activation are required for inhibiting the growth of ceritinib-resistant tumors. The occurrence of resistance was prevented all together when the tumors were treated with ceritinib in combination with CGM097 from the start (grey curve in *Figure 5H*), similar to the results shown in *Figure 3A*. Taken together, these data strongly suggest that combined inhibition of ALK and MDM2 in *TP53* wild-type neuroblastoma with ALK aberrations may provide an effective therapeutic modality capable of preventing as well as overcoming the resistance observed with the ALK inhibitor monotherapy.

## Discussion

Although the identification of activating *ALK* mutations and the demonstration of the oncogenic role of mutant ALK have established ALK as a therapeutic target in neuroblastoma, clinical studies of crizotinib and ceritinib in pediatric cancers have shown that patients with *ALK*-mutated neuroblastoma responded less favorably than pediatric patients with *ALK*-rearranged tumors, such as ALCL and IMT (*Mossé et al., 2013*; *Birgit Geoerger et al., 2015*). In the rare cases where the neuroblastoma patients had objective responses, their tumor cells expressed the ALK R1275Q mutation. Previous preclinical work has shown that crizotinib inhibited growth of neuroblastoma xenografts expressing ALK R1275Q or amplified wild-type ALK, but failed to inhibit growth of xenografts harboring F1174L-mutated ALK (*Bresler et al., 2011*). ALK F1174 mutations arose as acquired resistance mutations in oncogenic ALK fusion proteins during crizotinib and ceritinib treatments (*Friboulet et al., 2014*; *Sasaki et al., 2010*), suggesting that F1174L may cause primary resistance to certain ALK inhibitors in neuroblastoma. PF06463922 is a highly potent ALK inhibitor that is active against a range of ALK mutations, including F1174L. It has single-agent antitumor activity in neuroblastoma xenografts that express the three most common *ALK* mutations – F1174L, R1275Q and F1245C (*Infarinato et al., 2016*), providing the rationale to test more potent ALK inhibitors in clinic. However, there are still limitations likely associated with treating *ALK*-aberrant neuroblastoma with single-agent ALK inhibitors, including the newer generation inhibitors. In the clinical studies of crizotinib and ceritinib in pediatric patients with *ALK*-driven neuroblastoma, most tumors harboring R1275Q-mutated ALK did not respond to the treatments even although cells with the R1275Q mutation are sensitive to both drugs. In this preclinical study, consistent with the observations made by other groups (*Infarinato et al., 2016*), we found that PF06463922 treatment only led to 25% to 50% inhibition in cell lines expressing the ALK F1174L mutation (*Figure 1A*). Tumor cell plasticity may

render the remaining cells resistant to PF06463922 and eventually lead to recurrences during long-term treatments, which is the typical case with single-agent therapies. Multiple epidemiological studies on the distribution of *ALK* mutations in relation to *MYCN* status and other genomic parameters have revealed the co-occurrence of *ALK* mutations and *MYCN* amplification in neuroblastoma (*Bagci et al., 2012*; *De Brouwer et al., 2010*), indicating that additional oncogenic drivers exist other than ALK. Moreover, F1174L is significantly overrepresented in tumors with *MYCN* amplification, and almost all *ALK*-amplified tumors are also *MYCN*-amplified. All these findings suggest that the challenge for treating *ALK*-driven neuroblastomas is to overcome de novo resistance and combinatorial therapies are required to maximize the clinical benefit of ALK inhibition in neuroblastoma with ALK aberrations.

Previous studies have shown that ALK inhibition synergizes with chemotherapy or CDK4 inhibition in preclinical models of neuroblastoma (*Krytska et al., 2016*; *Wood et al., 2016*), and a clinical trial (NCT02780128) for testing the combination of ceritinib and ribociclib in *ALK*-mutant neuroblastoma patients is ongoing. In our study, several lines of evidence prompted us to assess the effectiveness of the combination of ceritinib and CGM097 in neuroblastoma models expressing activated ALK. First, activation of p53 signaling by MDM2 inhibitors provides a therapeutic approach for tumors that retain wild-type *TP53* (*Jeay et al., 2015*; *Weisberg et al., 2015*). In contrast to the high mutation rate of *TP53* in adult cancers, mutational inactivation of *TP53* occurs in less than 2% of neuroblastoma tumors at diagnosis (*Tweddle et al., 2003*). The rate of *TP53* mutations increases to 15% in relapsed neuroblastoma (*Carr-Wilkinson et al., 2010*). However, the p53 pathway is mainly inactivated via other mechanisms, such as MDM2 amplification and p14ARF impairment, in relapsed neuroblastoma (*Carr-Wilkinson et al., 2010*). These tumors should remain amenable to MDM2 inhibition. Second, MDM2 inhibition leads to increases in p53 levels and cell death in neuroblastoma through induction of cell cycle arrest and apoptosis. In agreement with our own work using CGM097 (*Figure 2A*), targeted disruption of the MDM2/p53 interaction by the small molecule MDM2 antagonists, such as nutlin-3 and RG7388, suppresses the proliferation of both chemoresistant and sensitive neuroblastoma cell lines with wild-type *TP53* (*Chen et al., 2015*; *Van Maerken et al., 2009*, *2006*). Finally, *ALK* mutations and *MYCN* amplification co-occur in a subset of neuroblastoma patients, emphasizing the need to target both oncogenes. Although it remains difficult to inhibit MYCN directly, disruption of the MDM2-p53 interaction by MDM2 inhibitors may provide an effective strategy for treating *MYCN*-amplified neuroblastoma. *MYCN*-amplified and *TP53* wild-type neuroblastoma cell lines show remarkable sensitivities to MDM2 antagonists in vitro (*Chen et al., 2015*; *Van Maerken et al., 2006*; *Gamble et al., 2012*). In addition, MDM2 contributes to tumorigenesis in a *MYCN*-driven transgenic mouse model for neuroblastoma, and its haploinsufficiency reduces MYCN-driven tumor incidence and inhibits tumor growth via p53-mediated apoptosis (*Chen et al., 2009*). In this study, we have demonstrated that treatment with ceritinib in combination with CGM097 synergistically inhibits the proliferation of *ALK*-mutated and *TP53* wild-type neuroblastoma cell lines in vitro and results in a shrinkage of tumor volumes below palpable detection in vivo, providing evidence for the effectiveness of combined inhibition of ALK and MDM2.

Resistance to targeted therapies remains a major problem in treatment of cancer patients. In most cases, tumors develop resistance to single-agent cancer therapeutics within a few months. To our knowledge, this is the first work that shows upregulation of MYCN acts as a mechanism by which ALK-driven neuroblastoma cells develop resistance to ALK inhibitors. We found that *ALK*- and *MYCN*-amplified NB-1 cells rapidly develop resistance to ceritinib even though they are extremely sensitive to ceritinib in culture with an IC50 value less than 50 nM. Several elegant studies in transgenic zebrafish and mouse models have already demonstrated synergistic effect of expressing activated ALK together with MYCN on inducing neuroblastoma (*Berry et al., 2012*; *Heukamp et al., 2012*; *Zhu et al., 2012*). Mutationally activated ALK may suppress the apoptotic response induced by MYCN, and prolonged treatment with ALK inhibitors can shift the equilibrium between pro-survival and pro-apoptotic pathways, which enables a subset of tumors cells to survive. However, additional stress, such as exposure to MDM inhibitors, can disrupt the balance established by ALK inhibition and lead the cells that become resistant to ALK inhibitors toward apoptosis. The detailed mechanisms underlying these effects remain to be determined. Given the observations that ALK F1174L occurs at a higher frequency in *MYCN*-amplified tumors and almost all *ALK*-amplified tumors are also *MYCN*-amplified, it is unlikely that durable responses can be achieved by targeting ALK alone. In this study, we have shown that CGM097, when combined with ceritinib, is able to prevent

and overcome the resistance conferred by MYCN upregulation through, at least in part, restoring the defects in p53 pathways and promoting apoptosis.

Some limitations of this study should be considered. Although ceritinib in combination with CGM097 induced complete and sustained tumor regression for the duration of the treatment in xenograft mouse models of neuroblastoma, tumor growth was observed 2 to 8 weeks after discontinuation of the treatment (*Figure 3A*). The recurrent tumors still responded to the combinatorial treatment after it was restarted (data not shown), suggesting that the treatment did not eliminate the residual disease. It will be interesting to test whether a more potent ALK inhibitor, such as PF06463922, in combination with CGM097 or other MDM2 antagonists can delay or prevent tumor regrowth after treatment termination. In addition, future work will be needed to assess the extent to which our findings can be extrapolated to xenograft models that express F1245C.

In summary, our results show that CGM097 significantly enhances the antitumor activity of ceritinib in neuroblastoma with aberrantly activated ALK by inducing a complementary set of anti-proliferative and pro-apoptotic signaling. We also find that neuroblastoma cell lines with *ALK* aberrations develop resistance to ALK inhibition as a result of MYCN upregulation. Induction of p53 by CGM097 in these ALK inhibition-resistant tumor cells restores their sensitivity. These findings support the clinical evaluation of combined inhibition of ALK and MDM2 as a therapeutic approach in *TP53* wild-type neuroblastoma with constitutively activated ALK.

## Materials and methods

### Cell lines and culture

The neuroblastoma cell lines used in this study and their *ALK*, *MYCN* and *TP53* genetic status are listed in *Table 1*. SH-SY5Y and IMR-32 were obtained from the American Type Culture Collection (ATCC), KELLY from Leibniz Institute DSMZ, NB-1643 from Children's Oncology Group (COG) Cell Culture and Xenograft Repository and NB-1 and KP-N-RT-BM-1 from JCRB Cell Bank. SH-SY5Y cells were cultured in a 1:1 mixture of EMEM and F12 Medium supplemented with 10% Fetal Bovine Serum (FBS), KELLY and NB-1 cells in RPMI-1640 supplemented with 10% FBS, IMR-32 cells in EMEM supplemented with 10% FBS, NB-1643 cells in IMDM supplemented with 20% FBS and 1X ITS (5 µg/mL insulin, 5 µg/mL transferrin and 5 ng/mL selenous acid) and KP-N-RT-BM-1 cells in RPMI-1640 supplemented with 15% FBS. The NB-1R cell line was generated by culturing NB-1 cells in 0.2 µM ceritinib in vitro for 3 weeks, and maintained in RPMI-1640 supplemented with 10% FBS and 0.2 µM ceritinib. The SH-SY5YR and KP-N-RT-BM-1R cell lines were generated by culture the parental cells in 1 µM PF06463922 in vitro for 3 to 4 months. All cell lines were determined to be free of mycoplasma contamination by a PCR detection methodology performed at Idexx Radil (Columbia, MO). The identity of the cell lines was confirmed periodically throughout the study using SNP fingerprinting.

### Reagents

Ceritinib and CGM097 were synthesized in the Global Discovery Chemistry Department at NIBR (Novartis). For in vitro studies, ceritinib and CGM097 were dissolved in 100% dimethyl sulfoxide (DMSO) at concentrations of 5 mM and 10 mM, respectively, and stored at −20°C until use. For in vivo studies, ceritinib was formulated in 0.5% CMC and 0.5% Tween 80 in water, and CGM097 was dissolved in 0.5% HPMC. Antibodies against phospho and total ALK, ERK, AKT, total p21, p27, PUMA, BIM and PARP were obtained from Cell Signaling, p53 from Santa Cruz Biotechnology, MYCN from Proteintech Group and MDM2 and GAPDH from EMD Millipore.

### Cell viability and proliferation assays

To determine the effects of compounds on cell proliferation, cells were plated in 96-well plates at a density of 5000–10,000 cells per well to achieve 70% of confluence in a volume of 100 µL, and grown for 24 hr prior to treatment. Cells were then treated with DMSO or compounds at concentrations ranging from 1.5 nM to 10 µM (threefold dilutions). After 6 days, cell proliferation was measured using the CellTiter-Glo luminescent cell viability assay (Promega, Madison, WI) and Victor4 plate reader (Perkin Elmer, Waltham, MA). Percent inhibition was calculated relative to DMSO signal, and all results shown are the mean of triplicate measurements.

For synergy analyses of drug combinations, 5000–10,000 cells suspended in 100 µl media were plated in triplicate into each well of 96-well tissue culture plates and grown for 24 hr prior to treatments. Ceritinib and CGM097 at different concentrations defined by a dose matrix were added to the cells such that all pair-wise combinations as well as the single agents were represented. Cells were incubated for 72 hr following compound addition, and cell viability was measured using the CellTiter-Glo luminescent cell viability assay (Promega) and Victor4 plate reader (Perkin Elmer). Isobolograms and combination indices were determined as described by Lehar et al. (*Lehár et al., 2009*).

## Western blot analysis

Cells were seeded into six-well plates at 50% of confluence, allowed to attach for 16 hr and treated. Cells were lysed in Cell Lysis Buffer (Cell Signaling Technology) containing Protease Inhibitor Cocktail (Roche Life Science, Indianapolis, IN) and PhosSTOP Phosphatase Inhibitor Cocktail (Roche). Lysates were centrifuged at 18,000 X g for 10 min at 4°C, and the supernatant was loaded to 4–12% Bis-Tris gradient gels (Invitrogen, Carlsbad, CA). The subsequent procedures were performed according to the recommendations of the antibody manufacturers.

## Silencing of MYCN

Cells were plated into six-well plates at 50% of confluence and allowed to attach for 16 hr. On the next day, cells were incubated with SMARTvector inducible lentiviral shRNA particles targeting *MYCN* or non-targeting control shRNA particles (Dharmacon, Lafayette, CO). The sequences targeted by these shRNAs are as follows: control, 5'-TGGTTTACATGTTGTGTGA-3'; MYCN, 5'-TAGG TATGAACTTCCAGTC-3'. Medium was replaced with fresh medium containing 1 µg/ml of puromycin after 24 hr, and selection was completed after 2 weeks. Transduced cells were maintained in RPMI-1640 supplement with 10% Tet System Approved FBS (Clontech, Mountain View, CA) and 1 µg/ml of puromycin. Expression of shRNAs was induced by 100 ng/ml or 1000 ng/ml of doxycycline in growth medium.

## RNA sequencing and data processing

RNA sequencing and data processing were conducted as described elsewhere (*Li et al., 2015*). Briefly, 0.5 µg of high-quality total RNA extracted from each fresh frozen tumor tissue sample was used to construct a sequencing library with the Illumina TruSeq RNA Sample Prep Kit. The libraries were sequenced on an Illumina HiSeq 2500, and transcriptomic alignments and gene expression quantification were performed using Array Studio 6.0 (OmicSoft, Cary, NC). The RNA-Seq data accession number in GEO is GSE82143.

## Xenograft studies

All in vivo studies were reviewed and approved by the Novartis Institutes of Biomedical Research Institutional Animal Care and Use Committee (IACUC) in accordance with applicable local, state, and federal regulations. The NB-1, NB-1643 and IMR-32 cells were harvested during exponential growth. Each athymic female nude mouse (Harlan Laboratories, Indianapolis, IN) was inoculated subcutaneously in the upper right flank with $2 \times 10^6$ NB-1 or IMR-32 cells suspended in 0.2 ml cold PBS. The severe combined immunodeficiency (SCID) mice were used to propagate NB-1643 xenograft tumors by injecting $1 \times 10^7$ NB-1643 cells suspended in a 1:1 mixture of cold PBS and Matrigel (BD Biosciences, San Jose, CA) subcutaneously into each animal. The development of SH-SY5Y xenograft tumors comprised two steps. In the first step, $1 \times 10^7$ SH-SY5Y cells harvested during exponential growth were suspended in a 1:1 mixture of cold PBS and Matrigel in a total volume of 0.2 ml, and injected subcutaneously into the upper right flank of nude mice. Tumors established in this step were collected and fragmented. Tumor fragments were then implanted into the upper right flank of nude mice. Tumor volumes were monitored with calipers twice per week. When tumor volume reached approximately 200 mm³, mice were randomized (n = 4 or 5 per group) and orally administered vehicle, 50 mg/kg ceritinib, 50 mg/kg CGM097 or 50 mg/kg ceritinib plus 50 mg/kg CGM097 daily, respectively.

In the study to test if CGM097 alone or CGM097 in combination with ceritinib is able to overcome ceritinib resistance in vivo, mice-bearing ceritinib-resistant NB-1 tumors were randomized into

three groups (n = 5 per group) when the tumors progressed under ceritinib treatment and the volume of ceritinib-resistant tumors reached 500 mm$^3$. In the first group, ceritinib single-agent treatment was terminated, and mice were treated with CGM097 at 50 mg/kg daily for 5 days. In the second group, ceritinib single-agent treatment was terminated, and mice were treated with CGM097 at 50 mg/kg daily for 5 days, and then CGM097 at 50 mg/kg plus ceritinib at 50 mg/kg daily for 43 days. In the third group, ceritinib single-agent treatment was terminated, and mice were treated with CGM097 at 50 mg/kg plus ceritinib at 50 mg/kg daily for 48 days.

## Statistical analysis

Values for the cell proliferation assays are expressed as means ± SEM. Statistically significant differences between the means were evaluated using ANOVA followed by Student's t-test. The p-values of multiple comparisons were corrected by the Bonferroni method.

## Acknowledgements

The NB-1643 cell line was provided by the Children's Oncology Group Cell Culture/Xenograft Repository.

## Additional information

### Funding

| Funder | Author |
| --- | --- |
| Novartis | Fang Li |

The research was funded by Novartis, Inc., where all authors were employees at the time the study was conducted. The authors declare no other competing financial interests.

### Author contributions

HQW, EH, Data curation, Formal analysis, Investigation, Methodology, Writing—original draft, Writing—review and editing; XL, JL, YC, Formal analysis, Investigation, Methodology; DPR, DAR, NT, SK, Investigation, Methodology; SJ, JUW, NL, JAW, Resources, Project administration; WRS, Resources, Project administration, Writing—review and editing; AH, Resources, Investigation, Project administration; FL, Conceptualization, Resources, Data curation, Software, Formal analysis, Supervision, Validation, Investigation, Visualization, Methodology, Writing—original draft, Project administration, Writing—review and editing

### Author ORCIDs

Fang Li, http://orcid.org/0000-0003-0497-4200

### Ethics

Animal experimentation: All in vivo studies were reviewed and approved by the Novartis Institutes of Biomedical Research Institutional Animal Care and Use Committee (IACUC) in accordance with applicable local, state, and federal regulations. If needed, a letter from the IACUC Chair can be provided to confirm that all in vivo studies were reviewed and approved by the Novartis IACUC. Below is the contact of the Novartis IACUC Chair. CeCe Brotchie-Fine, MA, CPIA Manager, Animal Welfare Compliance IACUC Chair & Animal Welfare Officer T +1 617 871 5064 M+1 617 834 4784 Email: Candice.brotchie-fine@novartis.com Novartis Institutes for BioMedical Research, Inc. 700 Main Street, 460 A Cambridge, MA 02139 USA

## Additional files

### Supplementary files

• Supplementary file 1. Gene expression levels in NB-1 xenograft tumors. NB-1 tumors that progressed under treatment with 50 mg/kg ceritinib and NB-1 tumors treated with vehicle as shown in

*Figure 3A* were analyzed by RNA-Seq. Gene expression was quantified by FPKM. FPKM, Fragments Per Kilobase of transcript per Million fragments mapped.

• Source code 1. R code for the statistical analysis shown in *Figure 5D*.

• Source code 2. R code for the statistical analysis shown in *Figure 5G*.

• Source code 3. R code for the statitical analysis shown in *Figure 5—figure supplement 1*.

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
