## [Decision Letter]

Thank you for submitting your article "Combined ALK and MDM2 inhibition increases antitumor activity and overcomes resistance in ALK mutant neuroblastoma" for consideration by *eLife*. Your article has been favorably evaluated by Tony Hunter (Senior Editor) and three reviewers, one of whom, Chi Van Dang, is a member of our Board of Reviewing Editors.

The reviewers have discussed the reviews with one another and the Reviewing Editor has drafted this decision to help you prepare a response to the significant concerns raised before a formal a revised manuscript is invited.

Summary:

Wang and colleagues determine that combination therapy of ALK inhibitor ceritinib with MDM2 inhibitor CGM097 cooperate in vitro and in vivo, and promote apoptosis in vitro ALK mutant/WT TP53 neuroblastoma cell lines. Combination therapy caused regression and increased survival in neuroblastoma xenograft models derived from two of these lines. Additionally, they report that ceritinib resistance may be caused by MYCN upregulation, and can be rescued by combination therapy. This work is of interest in the field given the challenges of inhibiting full-length mutant ALK in neuroblastoma. The major weakness is the limited in vitro and in vivo testing in models not fully representative of ALK alterations identified in patients, and the lack of data in ALK wild-type models to support this combination in a biomarker-driven population. While the mechanism for synergy by activation of non-overlapping pro-apoptotic and anti-proliferative proteins is well-supported, the mechanism for efficacy of CGM097 and combination therapy for ceritinib-resistant tumor lines is less clear, with the connection to MYCN relatively tenuous.

Essential revisions:

1) The cell line and xenograft sample set is insufficient given the breadth of publications in this field, and should be expanded in order to provide robust rationale for dual targeting of ALK and MDM2 in ALK-driven neuroblastoma models. The 4 cell lines screened (1 with ALK amplification and 3 with the same ALK F1174L mutation) are not representative of all 3 hot spot mutations found in patient samples, and no ALK wild-type models are evaluated in vitro or in vivo- this is problematic. A model harboring an R1275Q mutation should be tested, since this represents the most common ALK mutation in neuroblastoma and it is even referenced by the authors (Introduction, fourth paragraph) that the clinical responses reported in patients with this crizotinib/ceritinib-sensitive mutation is underwhelming. The in vivo experiments should also be expanded to models harboring the R1275Q mutation (and F1245), and would be more robust if the combination was tested in patient-derived xenografts as opposed to conventional xenografts generated from the cell lines. Several recent publications focused on ALK inhibition strategies in neuroblastoma show persuasive activity in PDX models, including compelling data in several such models with single agent PF06463922. The combination data with ceritinib/CGM907 presented here is limited and not convincingly superior.

2) Why was CGM097 used as opposed to HDM201, which is the compound in clinical trials?

3) The authors comment that "ALK mutations are frequently observed in MYCN-amplified neuroblastomas" and again reference the "co-occurrence of ALK mutations and MYCN amplification" in the Discussion. Important to note that in the largest series reporting the frequency and spectrum of ALK mutations in neuroblastoma (Bresler et al., Cancer Cell 2014), ALK mutations were found in 10.9% of MYCN-amplified tumors versus 7.2% of those without MYCN amplification (not a major difference). Additionally, the Discussion states that "ALK mutations and MYCN amplification co-occur in a subset of neuroblastoma patients with the poorest clinical outcome…," – this is not accurate. While patients with both amplified MYCN and F1174-mutated ALK had a significantly worse event-free survival than patients with neither, patient outcome did not differ significantly according to the location of the mutation in any analysis done in the Bresler et al. paper reporting on close to 1600 patients.

4) Despite compelling initial anti-tumor activity in the 2 in vivo models tested, tumors recurred shortly after discontinuation of therapy. In a recent paper by Krytska et al. (CCR 2015), crizotinib combined with standard cytotoxic agents sensitized ALK-mutant neuroblastoma xenografts with wild-type p53. When this combination was evaluated in SY5Y xenografts (same model tested with ceritinib/CGM907), known to be crizotinib-resistant, sustained complete tumor regressions were reported and maintained for 24-weeks post cessation of treatment. No reference is made to this paper and it is not clear how or if the ceritinib/CGM907 combination could be complementary or superior.

5) in vitro signaling data (Figure 2) do not persuasively show that dual targeting of ALK and MDM2 induced further activation of p53 or MDM2 compared to single agent MDM2 inhibition with CGM907; nor does the combination convincingly cause a significant increase of cleaved PARP, as noted by the authors. Additionally, the immunoblotting does not appear to show more profound downregulation of pERK and pAKT with the combination. Investigators should look for on-target activity in vivo, as kinetics of response are likely to differ.

6) The experiment showing that sensitivity to ceritinib in NB-1R could be restored by knocking down the expression of MYCN was elegant and suggests a role of MYCN in this model of resistance. It would be informative to assess whether MYCN downregulation (genetically and pharmacologically) impacts P-ALK levels.

7) In Figure 4, the authors suggest that p53 signaling might be impaired in the NB-1R cells. However, this suggestion does not correlate with higher sensitivity of to CGM097 in these cells compared to the sensitive counterpart cells (NB-1) – Figure 4. Also in Figure 4, p53 levels are very similar in both cell lines and do not correlate with the differences seen in p21 levels. Is it possible that p21 levels in these cells are being modulated by MYCN instead, as has previously been suggested (Iraci et al., 2011)?

8) Similarly to data shown in Figure 2 in the sensitive cells including NB-1, the combination of ceritinib and CGM097 in the resistant NB-1R cells does not convincingly increase levels of p53-related proteins compared to CGM097 single agent alone (Figure 4), overall suggesting that, at least in vitro, this combination is effective through other non-p53 mediated pathways. Immunoblotting analysis of tumors from in vivo experiments would be worth investigating.

9) Does cell death induction shown in Figure 2 and Figure 4 translate to induction of apoptosis in treated tumors in vivo? Can apoptosis analysis be added to Figure 2 and Figure 4?

10) The authors conclude that an increase in MYCN expression causes resistance to ceritinib in the NB-1 cells and CGM097 restores sensitivity. The MYCN shRNA experiment in Figure 4 shows restoration of ceritinib sensitivity in a single cell line. Can this be extended to support generality in additional lines, and does overexpression of MYCN in sensitive cells drive resistance?

11) Ceritinib appears to decrease the levels of p53, MDM2, and p21 levels in Figure 2. Do the authors have insight into this observation?

12) In Figure 4, where NB-1R cells are treated with monotherapy and combination therapy, are levels of MYCN restored to NB-1 parental levels? The blot for p-Alk here is poor quality and difficult to evaluate.

13) The RNA-seq data in Figure 3 represent the tumors that progressed under treatment with 50 mg/kg ceritinib and NB-1 tumors treated with vehicle. How many mice does this represent, and is this RNA-seq data an average? Can these details be more clearly stated? Additionally, while the genes displayed are specifically ribosomal subunit genes and other translation-related genes, what was the fold change of MYCN detected by their RNA-seq experiment? Do these data support data in Figure 3.

14) The authors suggest that the p53 axis becomes disrupted in the NB-1R cells, and also show these cells strongly decrease expression of p21, and that this confers resistance since treatment with CGM097 restores sensitivity. Are NB-1R cells less sensitive to etoposide treatment?

[Editors' note: further revisions were requested prior to acceptance, as described below.]

Thank you for resubmitting your work entitled "Combined ALK and MDM2 inhibition increases antitumor activity and overcomes resistance in human ALK mutant neuroblastoma cell lines and xenograft models" for further consideration at *eLife*. Your revised article has been favorably evaluated by Tony Hunter as the Senior Editor, Chi Dang as the Reviewing Editor and two reviewers.

The manuscript has been improved and responsive to the reviewers' comments, but there are some remaining issues that need to be addressed textually before acceptance. Please address the nuances of your findings in the Discussion or where applicable and submit the revision with altered text responding the key points of the following issues:

1) in vivo data shown in Figure 3 in ALKi-naïve cell line xenografts harboring varying ALK status: While antitumor activity with dual targeting of ALK and MDM2 is compelling, it does not appear superior to several already published data (crizotinib plus chemotherapy; ceritinib plus ribociclib), nor are the data convincingly superior to published data for lorlatinib alone in some of the very same models (NB1643 and SY5Y). How does time to regrowth compare with other combinations tested, or with lorlatinib alone? The anti-tumor activity in NB1643 is not compelling. While there is no doubt that combination strategies are needed to delay or overcome acquired resistance, the challenge for ALK-driven neuroblastomas is overcoming de novo resistance. Authors need to put their work in context with the published literature for this disease- especially in their Discussion.

2) It is interesting that lorlatinib was used to generate resistance in in vitro models – why was 1 μM used to treat parental cells? This is a very high dose (and certainly would not be relevant clinically) for such a potent ALK inhibitor, and the resulting phenotype is unlikely to model what will happen in patients.

3) Would still prefer to see data with HDM201; at the very least, authors need to mention that HDM201 is in fact in phase 1 testing in neuroblastoma patients with p53 wild type status (NCT02780128), as is the combination of ceritinib and ribociclib for neuroblastoma patients with ALK-driven tumors. How do they envision moving this to the clinic? Are they suggesting that lorlatinib (a Pfizer drug) be combined with HDM201 (a Novartis drug)?

4) Need to change very first sentence in Introduction – "Neuroblastoma is the most common extracranial childhood solid tumor"- this exact phrase has been used hundreds of times.

5) Fourth paragraph in Introduction, as well as early on in the Discussion, the authors postulate why pediatric patients with NB did not respond to crizotinib or ceritinib – the authors need to be very careful here as the implications made about why patients with ALK-mutant neuroblastoma did not respond to direct ALK kinase inhibition with crizotinib or ceritinib is not accurate. These are heavily pretreated patients whose tumor genome evolves dramatically under the selective pressure of multimodal therapy for high-risk neuroblastoma and selects for therapy resistance irrespective of the presence of an oncogenic driver such as ALK, and testing targeted therapies in this context is unlikely to reflect the therapy-naïve patients at diagnosis who are most likely to benefit from incorporation of ALK inhibition. In addition, I am not aware that the phase 1 study of ceritinib has been published.

6) A lot of data were generated in the NB1 model – important to put into context that ALK amplification is very rare in these patients (2% overall).

7) Discussion (fourth paragraph) – studying this combination in PDX models is likely more relevant than transgenic models.

8) Mechanism of synergy still not clear. Given the role of p53 in modulating cell cycle arrest combined with the finding that ceritinib alone induces P-Rb dowregulation in models of neuroblastoma (Wood A. et al., 2016), it would be informative to assess levels of this major cell cycle modulator and potential G1 arrest in the combination-treated cells versus single agent. For the resistance model, the authors suggest that the MYCN and p21 axis cannot explain sensitivity to the combination. Nevertheless, the levels of key modulators that induce p53 signaling are not shown. Also, if the p53 signaling responsible for regulating p21 is impaired in NB-1R, molecules downstream of p21 such as P-Rb, cyclins etc. should be assessed. Last, given that MYCN is a pivotal regulator of the cell cycle, it would be interesting to perform a cell cycle analysis in these cells.

---

## [Author Response]

*Essential revisions:*

*1) The cell line and xenograft sample set is insufficient given the breadth of publications in this field, and should be expanded in order to provide robust rationale for dual targeting of ALK and MDM2 in ALK-driven neuroblastoma models. The 4 cell lines screened (1 with ALK amplification and 3 with the same ALK F1174L mutation) are not representative of all 3 hot spot mutations found in patient samples, and no ALK wild-type models are evaluated* in vitro *or in vivo- this is problematic. A model harboring an R1275Q mutation should be tested, since this represents the most common ALK mutation in neuroblastoma and it is even referenced by the authors (Introduction, fourth paragraph) that the clinical responses reported in patients with this crizotinib/ceritinib-sensitive mutation is underwhelming. The in vivo experiments should also be expanded to models harboring the R1275Q mutation (and F1245), and would be more robust if the combination was tested in patient-derived xenografts as opposed to conventional xenografts generated from the cell lines. Several recent publications focused on ALK inhibition strategies in neuroblastoma show persuasive activity in PDX models, including compelling data in several such models with single agent PF06463922. The combination data with ceritinib/CGM907 presented here is limited and not convincingly superior.*

We agree with the reviewers that additional neuroblastoma models which contain different *ALK* hotspot mutations should be tested. We have established collaboration with the Genomics Institute of the Novartis Research Foundation to test the ceritinib and CGM097 combination in NB-1643 xenograft tumors. NB-1643 cells contain the ALK R1275Q mutation, *MYCN* amplification and wild-type TP53. We will also test this combination in vitroand in vivo in a neuroblastoma model which contains wild-type ALK. When work is completed, our revised manuscript will describe a diverse set of neuroblastoma models including those that contain *ALK* amplification, two different hot-spot ALK mutations (F1174L and R1275Q) and wild-type ALK.

We do not have access to neuroblastoma cell lines that contain the ALK F1245 mutations and neuroblastoma PDX models. We are aware that these models exist in a few labs in academic institutions. However, it usually takes at least six months to establish collaborations between external institutions and us, assuming that the proposal for the collaboration can be approved. Therefore, it is not feasible for us to include additional models, beyond the ones proposed, in a reasonable time frame.

We agree with the reviewers that PF06463922 is remarkably effective in xenograft models of neuroblastoma expressing the common *ALK* mutations. However, resistance to single-agent targeted therapies remains a major problem in treatment of cancer patients. Recently, we have established a cell line that shows significantly decreased sensitivity to PF06463922 as compared to the parental SH-SY5Y cell line. We have demonstrated marked increase of MYCN expression in PF06463922-resistant SH-SY5Y cells, and will test whether CGM097 can restore the sensitivity of these cells to PF06463922. In this study, we underscore the notion that concurrent inhibition of ALK and MDM2 provides an improved therapy compared to treatments with single-agent ALK inhibitors. We are hopeful that more potent ALK inhibitors, such as PF06463922, in combination with next-generation MDM2 antagonists can not only induce complete remission in *ALK*-mutated neuroblastoma but prevent tumor recurrence.

*2) Why was CGM097 used as opposed to HDM201, which is the compound in clinical trials?*

CGM097 is indeed also being tested in clinical trials. A Phase I dose escalation study of CGM097 in adult patients with selected advanced solid tumors with wild-type p53 is ongoing (NCT01760525). CGM097 and HDM201 share the same mechanism of action. Tested in several hundred cell lines of various tumor types, both compounds selectively inhibit the growth of the cells expressing wild-type p53. In these tumor cell lines, both compounds activate downstream p53 effector pathways that decrease cell proliferation and increase apoptosis. HDM201 is in fact a more potent inhibitor. However, HDM201 has a very fast clearance in mice. Given that the mechanism of action is identical and pharmacokinetic properties in the mouse are more desirable with CGM097, we feel that for the purposes of this study and for the conclusions being made, the use of CGM097 is just as adequate as it would be with HDM201. Moreover, our drug exposure studies of CGM097 and ceritinib when used in combination show no drug-drug interaction, further justifying the use of CGM097 for these combination studies.

*3) The authors comment that "ALK mutations are frequently observed in MYCN-amplified neuroblastomas" and again reference the "co-occurrence of ALK mutations and MYCN amplification" in the Discussion. Important to note that in the largest series reporting the frequency and spectrum of ALK mutations in neuroblastoma (Bresler et al., Cancer Cell 2014), ALK mutations were found in 10.9% of MYCN-amplified tumors versus 7.2% of those without MYCN amplification (not a major difference). Additionally, the Discussion states that "ALK mutations and MYCN amplification co-occur in a subset of neuroblastoma patients with the poorest clinical outcome…," – this is not accurate. While patients with both amplified MYCN and F1174-mutated ALK had a significantly worse event-free survival than patients with neither, patient outcome did not differ significantly according to the location of the mutation in any analysis done in the Bresler et al. paper reporting on close to 1600 patients.*

*We appreciate the point made by the reviewers. In the revised manuscript, we will make our comments on the co-occurrence of ALK mutations and MYCN amplification more specific. We will replace these sentences with the following ones: 1) ALK F1174 mutations occur in a high proportion of MYCN-amplified cases; 2) ALK mutations that co-occur with MYCN amplification were biased toward F1174 substitutions; 3) Patients with both the ALK F1174 mutations and MYCN amplification have a significantly worse event-free survival than patients with neither. We will cite the Bresler paper when we make these comments in the manuscript.*

*4) Despite compelling initial anti-tumor activity in the 2* in vivo *models tested, tumors recurred shortly after discontinuation of therapy. In a recent paper by Krytska et al. (CCR 2015), crizotinib combined with standard cytotoxic agents sensitized ALK-mutant neuroblastoma xenografts with wild-type p53. When this combination was evaluated in SY5Y xenografts (same model tested with ceritinib/CGM907), known to be crizotinib-resistant, sustained complete tumor regressions were reported and maintained for 24-weeks post cessation of treatment. No reference is made to this paper and it is not clear how or if the ceritinib/CGM907 combination could be complementary or superior.*

We will cite the paper published by Krytska et al. This paper describes another promising therapy for treating *ALK*-mutated neuroblastoma. However, it may not be straightforward to compare different treatments in different studies conducted at different labs. We foresee both of these therapies being tested in clinical trials, where the superiority or the lack of would be addressed.

*5) in vitro signaling data (Figure 2) do not persuasively show that dual targeting of ALK and MDM2 induced further activation of p53 or MDM2 compared to single agent MDM2 inhibition with CGM907; nor does the combination convincingly cause a significant increase of cleaved PARP, as noted by the authors. Additionally, the immunoblotting does not appear to show more profound downregulation of pERK and pAKT with the combination. Investigators should look for on-target activity in vivo, as kinetics of response are likely to differ.*

We report that combined inhibition of ALK using ceritinib and MDM2 using CGM097 induces a complementary set of anti-proliferative and pro-apoptotic proteins, which leads to greater apoptosis compared to each single agent alone. We do not think that the synergy is resulted from the enhanced target inhibition caused by the other compound. The increased levels of cleaved PARP in the three p53 wild-type cell lines treated with the combination is clear in Figure 2. Importantly, in the summary of the reviewers’ comment, the reviewers stated that “the mechanism for synergy by activation of non-overlapping pro-apoptotic and anti-proliferative proteins is well-supported”. However, we agree with the reviewer that we should assess the levels of target inhibition and apoptosis in vivo. We have already evaluated induction of apoptosis in treated NB-1 xenograft tumors and found that the combination induced greater apoptosis compared to the single-agent treatments, consistent with the observations made in vitro. The same experiment in SH-SY5Y xenograft tumors is ongoing. These results will be included in the revised manuscript.

*6) The experiment showing that sensitivity to ceritinib in NB-1R could be restored by knocking down the expression of MYCN was elegant and suggests a role of MYCN in this model of resistance. It would be informative to assess whether MYCN downregulation (genetically and pharmacologically) impacts P-ALK levels.*

We appreciate the reviewer’s positive comment. We will conduct the experiments suggested by the reviewer.

*7) In Figure 4, the authors suggest that p53 signaling might be impaired in the NB-1R cells. However, this suggestion does not correlate with higher sensitivity of to CGM097 in these cells compared to the sensitive counterpart cells (NB-1) – Figure 4. Also in Figure 4, p53 levels are very similar in both cell lines and do not correlate with the differences seen in p21 levels. Is it possible that p21 levels in these cells are being modulated by MYCN instead, as has previously been suggested (Iraci et al., 2011)?*

The reviewer is asking an excellent question. We will test whether p21 levels are modulated by MYCN in NB-1R cells transduced with lentiviral doxycycline-inducible vectors containing shRNA against *MYCN*. The p53 levels are very similar in NB-1 and NB-1R cell lines, but the p21 level is significantly lower in the NB-1R cell line than in the NB-1 cell line. Since p21 is one of the major p53 transcriptional target genes, we think that the p53 signaling responsible for regulating p21 expression might be impaired in the NB-1R cells.

We have demonstrated that increase in MYCN expression causes resistance to ceritinib in NB-1R cells because MYCN knockdown restores the sensitivity of NB-1R cells to ceritinib. MYC family proteins have two contradictory roles: promoting cell proliferation and sensitizing cells to apoptosis. The p14ARF-MDM2-p53 axis and its effectors keep the MYCN level in check. The defects in p53 signaling in NB-1R cells allow them to proliferate in the presence of significantly upregulated MYCN. Therefore, NB-1R cells are more sensitivity to CGM097 than the parental cells because they rely more on MYCN to proliferate and are more vulnerable to induced p53 signaling.

*8) Similarly to data shown in Figure 2 in the sensitive cells including NB-1, the combination of ceritinib and CGM097 in the resistant NB-1R cells does not convincingly increase levels of p53-related proteins compared to CGM097 single agent alone (Figure 4), overall suggesting that, at least in vitro, this combination is effective through other non-p53 mediated pathways. Immunoblotting analysis of tumors from in vivo experiments would be worth investigating.*

We cannot exclude the possibilities that CGM097 and its combination with ceritinib can have effects mediated through non-p53 pathways. However, ceritinib and CGM097 combination has no effect on KELLY, the p53-mutant cell line, suggesting that major contribution of CGM097 in this combination is the induction of p53 signaling. We are not suggesting that the synergy is the result of the combination being able to increase the levels of p53 more than CGM097 single agent alone. We provide data to describe the mechanism for synergy to be the activation of non-overlapping pro-apoptotic and anti-proliferative proteins from the combined ALK and MDM2 inhibition.

We agree with the reviewer that we should assess the levels of target inhibition and apoptosis in vivo. We have already evaluated induction of apoptosis in treated NB-1 xenograft tumors and found that the combination induced greater apoptosis compared to the single-agent treatments, consistent with the observations made in vitro. The same experiment in SH-SY5Y xenograft tumors is ongoing. These results will be included in the revised manuscript.

*9) Does cell death induction shown in Figure 2 and Figure 4 translate to induction of apoptosis in treated tumors in vivo? Can apoptosis analysis be added to Figure 2 and Figure 4?*

We thank the reviewer for this suggestion. We have already assessed induction of apoptosis in treated NB-1 xenograft tumors and found that the combination induced greater apoptosis compared to the single-agent treatments, consistent with the observations made in vitro. The same experiment in SH-SY5Y xenograft tumors is ongoing. These results will be included in the revised manuscript.

*10) The authors conclude that an increase in MYCN expression causes resistance to ceritinib in the NB-1 cells and CGM097 restores sensitivity. The MYCN shRNA experiment in Figure 4 shows restoration of ceritinib sensitivity in a single cell line. Can this be extended to support generality in additional lines, and does overexpression of MYCN in sensitive cells drive resistance?*

We do have additional lines to show that MYCN is upregulated in cells that are resistant to ALK inhibition. Recently, we have demonstrated markedly increased MYCN expression in PF06463922-resistant SH-SY5Y cells that harbor the ALK F1174L mutation, similar to the findings in ceritinib-resistant NB-1 cells. We chose PF06463922 in this cell line to achieve better ALK inhibition because SH-SY5Y contains the ALK F1174L mutation. We will test whether CGM097 can restore the sensitivity of these resistant cells to PF06463922. We are also treating KP-N-RT-BM-1 cells with PF06463922. If we can establish PF06463922-resistant KP-N-RT-BM-1 cells within the timeline we proposed for submitting the revised manuscript, we will conduct the same experiments in these cells, too. These results will be included in the revised manuscript.

We attempted to overexpress MYCN in NB-1 cells without success. We have preliminary results to show that upregulation of MYCN in the resistant cells is mainly caused by MYCN stabilization. This will be the focus of our future work.

*11) Ceritinib appears to decrease the levels of p53, MDM2, and p21 levels in Figure 2. Do the authors have insight into this observation?*

The reviewer is making an excellent comment. We made the same observation, but we do not have further insight into this observation. Downregulation of p53 by ALK inhibition may serve as a survival mechanism in *ALK*-mutated neuroblastoma cells treated with ALK inhibitors. Therefore, restoring or even increasing the level of p53 by MDM2 inhibitors can improve the efficacy of ALK inhibitors.

*12) In Figure 4, where NB-1R cells are treated with monotherapy and combination therapy, are levels of MYCN restored to NB-1 parental levels? The blot for p-Alk here is poor quality and difficult to evaluate.*

We will reblot p-ALK and assess the levels of MYCN as the reviewer suggested.

*13) The RNA-seq data in Figure 3 represent the tumors that progressed under treatment with 50 mg/kg ceritinib and NB-1 tumors treated with vehicle. How many mice does this represent, and is this RNA-seq data an average? Can these details be more clearly stated? Additionally, while the genes displayed are specifically ribosomal subunit genes and other translation-related genes, what was the fold change of MYCN detected by their RNA-seq experiment? Do these data support data in Figure 3.*

This represents two mice for each condition and the expression of each gene shown in Figure 3 is an average. We will provide the details in the revised manuscript. All the gene expression values are in the [Supplementary-material SD3-data]. The FPKM values of MYCN of control and ceritinib-resistant tumors are 529 and 638, respectively. Therefore, the increased protein level of MYCN is not due to the increased transcription of MYCN. We have preliminary data to show that upregulation of MYCN protein is caused by MYCN stabilization. MYCN is not a very stable protein, with a half-life less than one hour. The protein encoded by *MYCNOS (NCYM*) plays a role in stabilizing MYCN and its expression is increased more than 10 fold in the resistant tumors^1^. How MYCN is upregulated in ALK inhibitor resistant cells will be the focus of our future studies.

^1^ NCYM, a Cis-antisense gene of MYCN, encodes a de novo evolved protein that inhibits GSK3β resulting in the stabilization of MYCN in human neuroblastomas. Suenaga Y, et al. PLoS Genet. 2014

*14) The authors suggest that the p53 axis becomes disrupted in the NB-1R cells, and also show these cells strongly decrease expression of p21, and that this confers resistance since treatment with CGM097 restores sensitivity. Are NB-1R cells less sensitive to etoposide treatment?*

We do not think the decreased expression of p21 confers resistance. The increase in MYCN expression causes resistance to ceritinib in NB-1R cells because MYCN knockdown restores the sensitivity of NB-1R cells to ceritinib. MYC family proteins have two contradictory roles: promoting cell proliferation and sensitizing cells to apoptosis. The p14ARF-MDM2-p53 axis and its effectors keep the MYCN level in check. The defects in the p53 axis exemplified by the decreased expression of p21 in NB-1R cells allow them to proliferate in the presence of significantly upregulated MYCN.

[Editors' note: further revisions were requested prior to acceptance, as described below.]

*The manuscript has been improved and responsive to the reviewers' comments, but there are some remaining issues that need to be addressed textually before acceptance. Please address the nuances of your findings in the Discussion or where applicable and submit the revision with altered text responding the key points of the following issues:*

*1) in vivo data shown in Figure 3 in ALKi-naïve cell line xenografts harboring varying ALK status: While antitumor activity with dual targeting of ALK and MDM2 is compelling, it does not appear superior to several already published data (crizotinib plus chemotherapy; ceritinib plus ribociclib), nor are the data convincingly superior to published data for lorlatinib alone in some of the very same models (NB1643 and SY5Y). How does time to regrowth compare with other combinations tested, or with lorlatinib alone? The anti-tumor activity in NB1643 is not compelling. While there is no doubt that combination strategies are needed to delay or overcome acquired resistance, the challenge for ALK-driven neuroblastomas is overcoming de novo resistance. Authors need to put their work in context with the published literature for this disease- especially in their Discussion.*

We do agree with the reviewer that it is important to put our manuscript in the context of the published literature. We have cited the relevant literature in the Introduction and Discussion sections, and have put our manuscript in that context. However, we cannot make direct comparative statements between ours and the papers cited. This manuscript was not designed to conduct rigorous head to head comparisons. For the same reasons, we believe that the reviewer cannot claim superiority of other studies over ours either. Nonetheless, we do show marked antitumor activity of ceritinib in combination with CGM097 in these models, providing rationale for testing this combination in patients. Furthermore, previous publications do not describe mechanisms responsible for resistance to ALK inhibition nor do they provide tested suggestions to overcome the resistance. We however demonstrate that concomitant inhibition of MDM2 and ALK can overcome ALK inhibitor-resistance conferred by MYCN upregulation.

The reviewer suggests that the anti-tumor activity in NB-1643 xenografts is not compelling. We observed that ceritinib monotherapy only delayed the tumor growth and the tumor volume increased more than 2-fold compared to day 0. In contrast, the combination treatment resulted in nearly complete tumor regression (7% of the tumor volume at day 0). After we submitted the revised manuscript, the treatments were stopped, but we continue to observe the animals. We found sustained tumor regression in the combination-treated group. We have updated the NB-1643 results in Figure 3 to reflect this additional observation. We think that the anti-tumor activity of the ceritinib/CGM097 combination treatment in this model is even more striking.

We do agree with the reviewer that the challenge for *ALK*-driven neuroblastomas is overcoming de novo resistance. Our Figure 1 and Figure 3 speak to that point. To make this point clearer, and as the reviewer suggested, we added the following text in the Discussion: all these findings suggest that the challenge for treating *ALK*-driven neuroblastomas is to overcome de novo resistance and combinatorial therapies are required to maximize the clinical benefit of ALK inhibition in neuroblastoma with *ALK* aberrations (Discussion, end of first paragraph).

*2) It is interesting that lorlatinib was used to generate resistance in* in vitro *models – why was 1 μM used to treat parental cells? This is a very high dose (and certainly would not be relevant clinically) for such a potent ALK inhibitor, and the resulting phenotype is unlikely to model what will happen in patients.*

To develop lorlatinib-resistant SH-SY5Y and KP-N-RT-BM-1 cells, we wanted to choose a dose that achieves maximum inhibition of ALK while still retaining on-target selectivity (using suboptimal dosing could also result in phenotypes that are not modeling the relevant and complete inhibition of ALK). In our studies, the dose response curves showed no off-target killing in SH-SY5Y and KP-N-RT-BM-1 cells when lorlatinib was dosed at 1 µM (Figure 1), while at that same dose the strongest phospho-ALK inhibition was observed (Figure 1).

*3) Would still prefer to see data with HDM201; at the very least, authors need to mention that HDM201 is in fact in phase 1 testing in neuroblastoma patients with p53 wild type status (NCT02780128), as is the combination of ceritinib and ribociclib for neuroblastoma patients with ALK-driven tumors. How do they envision moving this to the clinic? Are they suggesting that lorlatinib (a Pfizer drug) be combined with HDM201 (a Novartis drug)?*

As the reviewer suggested, we have mentioned the clinical study of ceritinib and ribociclib combination in the Discussion section (second paragraph). However, the treatment of single agent HDM201 in *TP53* wild-type neuroblastoma is not addressing *ALK* mutants.

The following criticism from the reviewer: “Are they suggesting that lorlatinib (a Pfizer drug) be combined with HDM201 (a Novartis drug)?” we find to be off-base. The scientific merit of a paper does not hinge on the clinical candidacy. As such, numerous articles have been published in *eLife* and elsewhere with molecules that are not clinical candidates (for example consider publications with JQ1). Furthermore, it is beyond the scope of this scientific manuscript and this letter to provide justification of how two pharmaceutical companies could do business together and jointly bring drugs to the patients. However, we believe that our manuscript provides a strong rationale and a novel idea to test the efficacy of combined inhibition of ALK and MDM2 in *ALK*-mutant neuroblastoma.

*4) Need to change very first sentence in Introduction – "Neuroblastoma is the most common extracranial childhood solid tumor"- this exact phrase has been used hundreds of times.*

The first sentence was changed to “Neuroblastoma is a common pediatric solid tumor that arises from neural crest cells”.

*5) Fourth paragraph in Introduction, as well as early on in the Discussion, the authors postulate why pediatric patients with NB did not respond to crizotinib or ceritinib – the authors need to be very careful here as the implications made about why patients with ALK-mutant neuroblastoma did not respond to direct ALK kinase inhibition with crizotinib or ceritinib is not accurate. These are heavily pretreated patients whose tumor genome evolves dramatically under the selective pressure of multimodal therapy for high-risk neuroblastoma and selects for therapy resistance irrespective of the presence of an oncogenic driver such as ALK, and testing targeted therapies in this context is unlikely to reflect the therapy-naïve patients at diagnosis who are most likely to benefit from incorporation of ALK inhibition. In addition, I am not aware that the phase 1 study of ceritinib has been published.*

The reviewer raised an interesting hypothesis. In the Introduction section, we describe the lack of efficacy of ALK inhibitors in *ALK*-mutant neuroblastoma in the context of other ALK-driven cancers. Early phase clinical trials of crizotinib and ceritinib showed significant clinical activity in patients with non-small cell lung cancer (NSCLC), anaplastic large cell lymphoma (ALCL) and inflammatory myofibroblastic tumor (IMT) that harbor *ALK* rearrangements. These patients were also heavily pretreated with multimodal therapies.

The clinical study of ceritinib in pediatric cancer has been published. The citation is listed below and in the manuscript.

Birgit Geoerger, J.S., Christian M. Zwaan, Michela Casanova, Matthias Fischer, Lucas Moreno, Toby Trahair, Irene Jimenez, Hyoung Jin Kang, Alberto S. Pappo, Eric Schafer, Brian D. Weiss, Mary Ellen Healy, Ke Li, Tiffany Lin, Anthony Boral, Andrew DJ Pearson, Phase I study of ceritinib in pediatric patients (Pts) with malignancies harboring a genetic alteration in ALK (ALK^+^): Safety, pharmacokinetic (PK), and efficacy results. J Clin Oncol, 2015. 33(suppl): p. abstr 10005.

*6) A lot of data were generated in the NB1 model – important to put into context that ALK amplification is very rare in these patients (2% overall).*

Our revised manuscript describes a diverse set of neuroblastoma models including those that contain *ALK* amplification, two different hot-spot *ALK* mutations (F1174L and R1275Q) that account for 73% of ALK mutant neuroblastoma. The NB-1 cell line and xenografts were used as the major model for studying the resistance mechanism because it is the only model that is truly sensitive to ALK inhibition. We also included two additional models that contain the F1174L mutation for studying mechanism responsible for resistance to ALK inhibition. As the reviewer requested, we have added the frequency of ALK amplification in the Introduction section (second paragraph).

*7) Discussion (fourth paragraph) – studying this combination in PDX models is likely more relevant than transgenic models.*

Modified according to the reviewer’s suggestion.

*8) Mechanism of synergy still not clear. Given the role of p53 in modulating cell cycle arrest combined with the finding that ceritinib alone induces P-Rb dowregulation in models of neuroblastoma (Wood A. et al., 2016), it would be informative to assess levels of this major cell cycle modulator and potential G1 arrest in the combination-treated cells versus single agent. For the resistance model, the authors suggest that the MYCN and p21 axis cannot explain sensitivity to the combination. Nevertheless, the levels of key modulators that induce p53 signaling are not shown. Also, if the p53 signaling responsible for regulating p21 is impaired in NB-1R, molecules downstream of p21 such as P-Rb, cyclins etc. should be assessed. Last, given that MYCN is a pivotal regulator of the cell cycle, it would be interesting to perform a cell cycle analysis in these cells.*

In our manuscript, we indeed showed that CGM097 caused induction of p21 and PUMA, while ceritinib caused induction of p27 and Bim. Therefore, ceritinib and CGM097 induce a complementary, yet non-overlapping set of anti-proliferative and apoptosis stimulating molecules, which resulted in synergistic antitumor effects. It is well established that induction of p21 and p27 leads to cell cycle arrest, and that PUMA and Bim function through BH3-dependent mechanisms to induce apoptosis. It would be interesting to further differentiate and monitor temporal contributions of cell cycle arrest and apoptosis to the effectiveness of this combination in future studies.